**METHOD**

# Hybrid-hybrid correction of errors in long reads with HERO

Xiongbin Kang[1,2], Jialu Xu[1], Xiao Luo[1*] and Alexander Schönhuth[2*]

*Correspondence:
xluo@hnu.edu.cn;
aschoen@cebitec.uni-bielefeld.de

[1] College of Biology, Hunan University, Changsha, China
[2] Genome Data Science, Faculty of Technology, Bielefeld University, Bielefeld, Germany

## Abstract

Although generally superior, hybrid approaches for correcting errors in third-generation sequencing (TGS) reads, using next-generation sequencing (NGS) reads, mistake haplotype-specific variants for errors in polyploid and mixed samples. We suggest HERO, as the first "hybrid-hybrid" approach, to make use of both de Bruijn graphs and overlap graphs for optimal catering to the particular strengths of NGS and TGS reads. Extensive benchmarking experiments demonstrate that HERO improves indel and mismatch error rates by on average 65% (27∼95%) and 20% (4∼61%). Using HERO prior to genome assembly significantly improves the assemblies in the majority of the relevant categories.

**Keywords:** Correction of sequencing errors, Haplotype specific variation, Metagenome sequencing, Third-generation sequencing reads, Genome assembly

## Background

Third-generation sequencing (TGS) platforms have decisively promoted genomics research. Because TGS reads span from tens of thousands to a few million base pairs [1], their advantages are immediately obvious. TGS reads can bridge longer-range repetitive genomic regions (for example large tandem repeats [2]), and they make decisive contributions to sequencing centromeres [3], telomeres [4], and highly identical regions in multiple haplotypes [5]. TGS reads also support the accurate identification of complex structural variants [6], and they are the foundation for enhanced haplotype phasing [7], structural variant calling [8], and the assembly of (complex) genomes [9]. From a technical perspective, their advantages in comparison with the earlier (but for various good reasons still widely popular) next-generation sequencing (NGS) reads, which span only a few hundred base pairs, are to reduce the ambiguity in terms of their origin. This leads to statistically more certain scenarios when arranging the reads with each other.

The much reduced sequencing error rates of the short NGS reads, however, remain a major advantage over the long TGS reads. While the error rates affecting NGS reads range at less than 1% [10], error rates of the most popular, *inexpensive* classes of TGS

reads amount to 5–15% [1]. On the other hand, the considerably more expensive PacBio HiFi (including Revio HiFi) and ONT Q20+ reads exhibit error rates of about 1% (Q20) or even below (Q30), thanks to employing refined, elaborate sequencing protocols. In brief, their advantages in terms of error rates are offset by their costs (up to 3 times more expensive than PacBio CLR or regular ONT type reads [1, 11]), by their reduced length (approx. 3 times shorter than PacBio CLR reads or 3–5 time shorter than Oxford Nanopore (ONT) reads; see again, for example [1, 11]), and by the efforts required to handle their elaborate protocols, which imply further non-negligible computational investments.

Of further relevance, the spectrum of errors affecting TGS reads is dominated by artificial insertions and deletions (indels). This distinguishes them from NGS reads, which are predominantly affected by artificial substitutions. So, at any rate, TGS and NGS reads still complement each other to a degree that single classes of (TGS or NGS) reads can not compete with.

Notwithstanding analysis scenarios in which elevated error rates are not much of an issue, tasks such as de novo assembly, variant calling, or outlining high resolution intron-exon boundaries require enhanced base-level accuracy [12]. So, removal of errors from TGS reads prior to analysis is key in exactly the experimental scenarios where elevated read length is decisive.

The fact that TGS and NGS reads complement each other in terms of length and accuracy provides hope—when done right, combining their virtues yields reads that are the longest possible and of utmost accuracy. The fact that laboratories worldwide are equipped with NGS *and* TGS platforms supports the practical feasibility of the idea. For laboratories that operate at limited budgets, which represent the vast majority of laboratories worldwide, combining cheap versions of TGS with (the anyway very cheap) NGS is a perfectly viable option. Note that even for the rather few laboratories that run on elevated budgets, combining TGS with NGS may mean considerable advantages because of the elevated length of the cheaper versions of the TGS reads (depending on the particular platform, up to 3-5 times longer than PacBio HiFi, for example, see above), and possibly also the error rates, because hybrid corrected reads are even more accurate than the latest versions of HiFi or ONT reads by themselves. In other words, synthesizing the two classes of reads in an optimal way yields reads that excel in an overall comparison of currently available sequencing data: they are (by far) the longest and the most accurate. Investments in protocols that integrate both TGS and NGS reads therefore, apart from being less convenient, can considerably pay off both in terms of expenses and read quality.

*Hybrid error correction (HEC)* is the canonical idea that addresses the combined exploitation of the complementary advantages of TGS and NGS. Standard protocols of HEC either align NGS reads with TGS reads or vice versa. Then, they eliminate the errors in TGS reads by evaluating the alignment derived juxtapositions. This obvious idea is not novel at all; however, as always, the devil is in the details. Leading related examples are LoRDEC [13], FMLRC [14] and Ratatosk [15] (among various other approaches).

There is an additional level of complementarity beyond that of the properties of the reads: the computational paradigms that support the optimal processing and arrangement of the two types of reads are also complementary. While the organization and

analysis of NGS reads are optimally supported by k-mer-based data structures, organization and analysis of TGS reads decisively depends on capturing their interrelationships at their full length.

De Bruijn graphs (DBGs), as $k$-mer-based data structures, have proven to capture the essential information inherent to NGS reads on plenty of occasions. DBGs particularly cater to the short length of the reads and the often large volumes and the corresponding redundancies that NGS read sets bring along. In more detail, vertices in DBGs refer to $k$-mers while edges bridge $k$-mers if the $k - 1$-suffix of the first $k$-mer matches the $k - 1$-prefix of the second $k$-mer. Dealing with $k$-mers implies to decompose reads into the spectrum of $k$-mers they harbor. This works well as long as $k$, which is shorter than the read itself, is not too short in relation to the length of the read itself. Capturing NGS reads by their $k$-mers, and discarding the full length reads otherwise, leads to condensed representations of the read sets, because $k$-mers re-appear in multiple reads, which greatly reduces the redundancy of the read sets. Plenty of examples provide evidence of successful such practice [16–18].

On the other hand, as documented by recent developments, overlap-based data structures, such as multiple, full-length alignment of reads, or overlap graphs (OGs) where the latter arrange the overlaps in a way that supports assembly in particular, are superior because they preserve the sequential information of the reads at their full length. In fact, overlap-based data structures have experienced a renaissance precisely because of TGS technology. In some more detail, applying DBG-based techniques to TGS reads would annihilate their very advantages, because the $k$ for DBG-based $k$-mers is much shorter than the length of TGS reads. For example, OGs capture the information of the reads at their full length, because vertices in OGs refer to the full length reads, while edges connect vertices whose overlap is of sufficiently good quality in terms of length and matching statistics. Unsurprisingly, the OG paradigm has become the prevailing mode of operation for assembling TGS reads (see [19–24] for leading examples).

All HEC approaches so far developed rely on either DBG-based or overlap/multiple alignment (MA)-based structures: no approach synthesizes the paradigms. Here, we present the first tandem hybrid ("hybrid-hybrid") approach. To the best of our knowledge, combined and explicit usage of DBGs and MAs/OGs for long read error correction is a novelty.

We aim to harness the properties of NGS reads on the one hand, and to harness the properties of TGS reads, on the other hand, by employing both DBGs and MAs/OGs. The goal is to synthesize the two axes of hybridity into one encompassing and superior approach. The final outcome are clean long reads that are more accurate than reads resulting from applying only one of the paradigms, and neglecting the other.

### Related work

Long read correction strategies can generally be divided into self-correction methods and HEC methods [25]. Self-correction refers to correcting TGS reads without making use of additional information. Instead, one aligns the TGS reads with each other via their overlaps and derives template sequences from the alignments that are as accurate as possible. One then corrects the errors in the individual reads based on their (multiple) alignment with the template sequences. Among various others,

prominent self-correction approaches are PBcR [26], HGAP [27], Sprai [28], LoRMA [29], and VeChat [30]. Because low coverage corrupts the accuracy of the consensus, self-correction methods require sufficient read coverage. This entails two disadvantages. First, it is not feasible to correct errors when coverage is low [31, 32]. Secondly, MAs (or, respectively, the OGs immediately derived from the MAs), as the predominant data structure that underlies self-correction methods, are not well-behaved with respect to too large read volumes [30]. This narrows the spectrum of applications of self-correction methods considerably.

The second class of methods are the HEC methods themselves. As above-mentioned, HEC methods aim to correct errors in TGS reads using shorter, but more accurate NGS reads drawn from the same DNA sample. NGS is relatively inexpensive, and the accuracy of the resulting reads is high. Therefore, HEC is a particularly feasible strategy when aiming to rescue long reads in low-coverage regions or when drawn from low-abundance haplotypes [32]; note that self-correction methods particularly struggle with such scenarios. As mentioned earlier, HEC approaches can, in turn, be classified into two categories.

(1) *DBG-based HEC strategies* follow the idea to construct DBG-based indexes. Subsequent alignment of the long reads against the resulting DBGs (using the indexes) leads to identification and correction of the errors that they carry. Predominant such approaches, among others, are LoRDEC [13], FMLRC [14], and Ratatosk [15]. Of note, FMLRC has been reported to be superior in recent benchmark studies [31, 32]. Although reasonably effective, these methods tend to suffer from the general shortcomings of DBGs. (a) Because some long reads do not align with the DBG unambiguously in complex and/or repetitive regions, the affected reads remain uncorrected. (b) Because of the loss of information on genetic linkage of variants, DBGs get confused with haplotype/strain specific mutations. This implies that long reads often align with similar, but mistaken haplotypes, which leads to confusion when identifying errors. For example, they struggle to distinguish between pathogenic strains (see [33] for remarks).

(2) *MA-based HEC methods* align NGS reads directly against TGS reads. Subsequently, they temporarily discard the template and compute an MA of the NGS reads. The MA then serves as the basis for overlap-based structures, usually reflected by OGs. One can traverse the OGs to assemble the NGS reads into clean sequence. Eventually, the assembled clean sequence replaces the template, which virtually removes all errors from the template [31]. Despite preserving linkage information, this class of approaches has issues in its own right. First, in long repeated elements or very similar haplotypes/strains, the wrong short reads are aligned to the long reads, which leads to introduction of even more errors, often referred to as "overcorrection." In addition, the fact that long reads are predominantly affected by indel errors leads to mistaken alignments with the short reads, due to the usual issues induced by mistaken indel placements. Unsurprisingly, MA-based approaches have proven to be little effective in recent benchmark studies [25, 31].

For summarizing accounts on the particular advantages of DBGs on the one hand and MAs/OGs on the other hand, mostly referring to de novo assembly as the primary domain of application of HEC because de novo assembly crucially depends on the availability of clean, long reads, see also [34–36].

Notwithstanding the issues that remain—and which we would like to overcome—DBG-based HEC establishes the leading methodology. In contrast, usage of MAs in HEC has remained in its infancy. From this vantage point, one can alternatively interpret our approach as a novel way to leverage the advantages of overlap-based data structures (i.e., MAs and OGs) for HEC. To the best of our current understanding, this is only possible when making hybrid use of DBGs and MAs and the OGs derived from the MAs in particular. Note finally that while further development of MA-based HEC methodology appears imperative, the further development of DBG-based HEC methodology appears to yield only marginal improvements. This insight provided guidance for the development of our workflows: we focused on improvements with respect to MA-based methodology in particular.

### Summary of contributions

In this study, to the best of our knowledge:

1  We present the first approach that does not make use of only one, but of both DBGs and (full-length) MAs (and the OGs immediately derived from them) for correcting errors in long TGS reads.
2  Thereby, we establish the first HEC method that integrates and synthesizes the two complementary axes. It exploits DBGs on the one hand and overlap-based structures on the other hand for optimized treatment of the characteristic properties of NGS and TGS reads. It therefore is the first approach that is double hybrid ("hybrid-hybrid").
3  We establish the first method that explicitly addresses to correct errors in genomes whose TGS read coverage is low.
4  We establish the first method that explicitly addresses to correct errors in genomes that are characterized by multiple haplotypes that are very similar to each other, like different strains referring to identical species in metagenomes.

## Results

### Approach

We have designed and implemented HERO ([H]ybrid [E]rror co[R]recti[O]n), a novel approach that makes combined use of DBGs and MAs/OGs for the purposes of HEC of TGS reads. HERO is "hybrid-hybrid" insofar as it uses both NGS and TGS reads on the one hand, hybrid in terms of using reads with complementary properties, and both DBGs and MAs/OGs on the other hand, hybrid with respect to the employment of complementary data structures.

The foundation of HERO is the idea that aligning the short NGS reads with the long TGS reads prior to correction yields corrupted alignments because of the abundantly occurring indel artifacts in the TGS reads. This has exposed approaches that are exclusively (multiple) alignment (or even OG) based as inferior, which has blocked further

progress in terms of the usage of (multiple) alignments/OGs in HEC. When analyzing earlier related work further (such as most importantly LoRDEC [13], FMLRC [14], and Ratatosk [15]), it becomes evident that DBG-based approaches have the power to remove sufficiently many mistaken indels from the TGS reads to render NGS-TGS-read alignments sufficiently meaningful.

This motivates the strategy that we have implemented. First, we use DBGs for removing artificial indels from the TGS reads to a degree that does no longer disturb subsequent alignments. Second, we use overlap derived alignments for identifying mutations that characterize near-identical haplotypes. We recall that mistaking haplotype-specific mutations for errors is a primary source of issues when making use of DBGs alone.

Examining the prior approaches, this sequential protocol—first DBGs to make alignments work and second, now equipped with reliable aligments, MAs/OGs to take care of haplotype-specific, linked mutations—appears to be imperative. HERO implements this basically sequential strategy.

Note that this protocol can be extended to iterative schemes, where single elements of this protocol are repeatedly applied. To refine and optimize the correction, HERO also makes use of such iterative strategies. In this, the predominant principle is repeated application of DBGs followed by repeated application of MAs/OGs; see the "Workflow" section and the "Methods" section for further details.

As for the implementation of the different parts of the protocol, we found that for DBG-based pre-correction, employing known, heavily researched, so sufficiently optimized methodology lead to optimal results. In particular, we build on LoRDEC [13], FMLRC [14], and Ratatosk [15] for implementing the DBG-related parts of the workflow of HERO. To the best of our understanding, the protocols and workflow that we present are novel concepts.

Following DBG-based pre-correction, the MA/OG-based part that we present is methodologically novel also in its own right. We recall that research on MA (or even OG)-based approaches for HEC has been in its infancy. This motivates the substantial further developments of ours.

We have drawn our inspiration for the MA/OG-based part predominantly from haplotype-aware assembly approaches; see [9, 21, 37] for examples. The basic principle that underlies these strategies is to identify SNPs in the overlaps of the TGS reads. Without disturbing indel artifacts in the TGS-NGS-read alignments—removed thanks to the DBG part—identification of haplotype-specific SNPs is now possible.

After the identification of SNPs, HERO reconsiders the alignments of the NGS reads with the TGS reads themselves. It is now possible to phase the NGS reads by discarding mistaken and maintaining correct TGS-NGS-read alignments. The removal of mistaken TGS-NGS-alignments leads us to a superior level of correction, namely the level of correcting TGS reads relative to the phases (haplotypes/strains etc) they stem from. Enabling this level of correction means to have put an improved correction strategy into effect.

The advantages of our novel strategy become most evident in scenarios relating to DNA samples characterized by multiple haplotypes/strains/lineages. We will therefore focus on metagenomics as one of the envisioned primary domains of application of HERO, on the one hand, and on diploid/polyploid plant genomes, on the other hand.

Note in particular that HEC methods so far presented predominantly struggle with metagenome datasets or sequencing scenarios that are characterized by the issues inherent to polyploid or mixed samples.

From this point of view, we present the first HEC method applicable for metagenome sequencing data in particular and an HEC method that also caters to polyploid scenarios in a particularly favorable way. In a brief summary (see the "Results" section for details), HERO reduces indel and mismatch error content by on average 65% (indels) and 20% (mismatches) in the metagenome datasets in comparison with prior state-of-the-art approaches, when correcting errors in PacBio CLR and ONT reads (as the classes of TGS that yield the longest reads). These advantages are certainly substantial; for indels, as the major class of errors affecting TGS reads, they are arguably even drastic. Last but not the least, we will demonstrate that when using HERO prior to (strain-aware) metagenome assembly or haplotype-aware assembly of polyploid genomes, results become improved substantially as well: one experiences considerable advantages in terms of the accuracy and completeness of the draft genomes, as the two most important categories, in particular. This underscores the practical necessity to invest in refined, optimized error correction strategies.

In the following, in the "Workflow" section, we present the workflow of HERO. Subsequently, in the "Data and experimental setup" section, we discuss the data, the design of the experiments. In the subsequent subsections, we present the results that HERO achieved in comparison with the state-of-the-art HEC approaches.

### Workflow

In this section, we provide a high-level description of the workflow. See the "Methods" section for detailed descriptions of all methodical steps involved.

See Fig. 1 for an illustration of the overall workflow of HERO. As outlined above, HERO consists of two stages.

*The first stage* consists of employing well-established, high-quality DBG-based HEC tools such as Ratatosk [15], FMLRC [14], and LoRDEC [13]. Attempts of ours to achieve further progress in terms of the DBGs involved did not lead to further improvements. Therefore, we also refrain from providing further illustration but refer the reader to the publications just mentioned for details.

*The second stage* is the (multiple) alignment (and also overlap graph)-based correction, which is methodologically novel, and *consists of six major steps*. These six major steps are illustrated in more detail in Fig. 1. Numbers in the Fig. 1 refer to the number of steps as explained here.

*Step 1* consists of computing minimizer-based all-vs-all overlaps between each NGS read and each TGS reads on the other hand. For this, we make use of Minimap2 [19], as the currently leading tool for that purpose.

Steps 2–5 reflect the technical core of our read phasing and MA/OG-based error correction procedure. See Fig. 1 for an illustration.

In *step 2*, HERO picks one TGS reads as the target read and computes an NGS read alignment pile that consists of all short reads that overlap the target read. The NGS read alignment pile may include substantial amounts of NGS reads that originate from haplotypes that differ from the one the target read was drawn from. Keeping such NGS reads

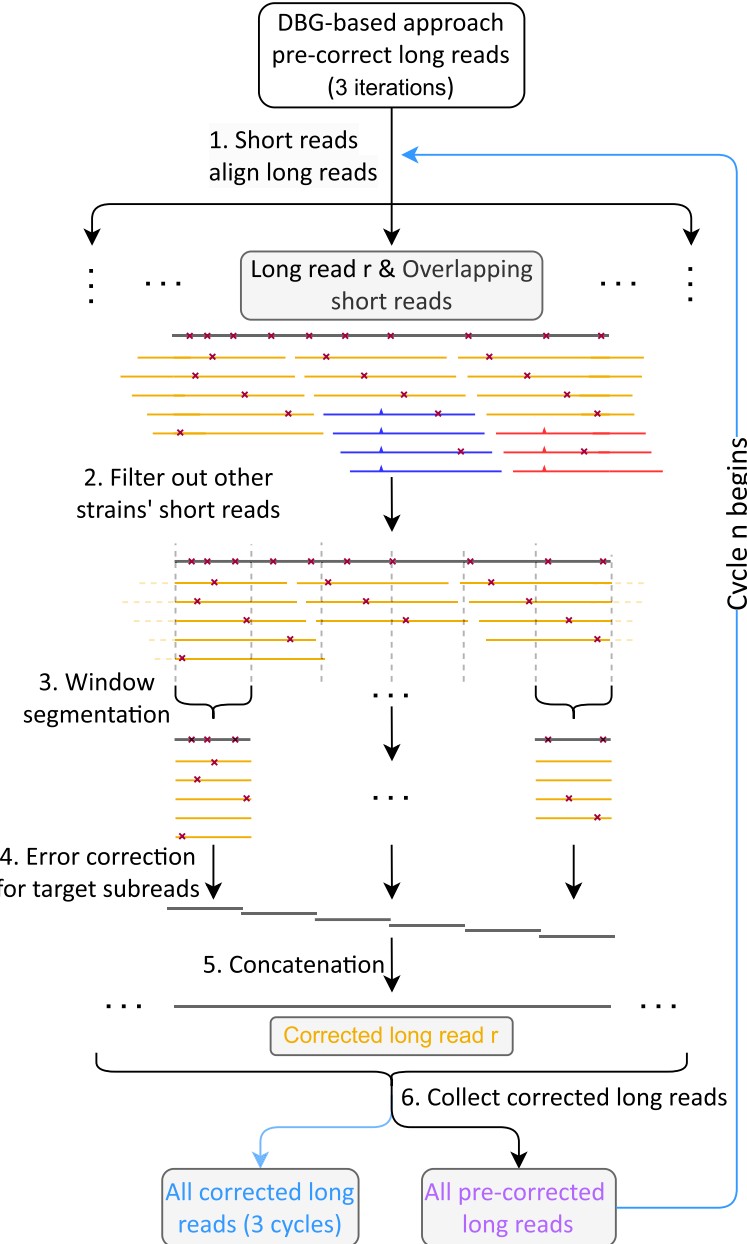

**Fig. 1** The workflow of HERO. Black: long read *r* to be corrected. Yellow, blue, red: short reads, different colors indicate different haplotypes. Red crosses indicate errors. Tics (here: blue, red) indicate haplotype-specific variants

in the pile would lead to "overcorrection" of the true mutations contained in them. To appropriately deal with the problem, we phase NGS reads, and filter out those that stem from other strains, based on the SNP information that one obtains from the overlap computations. We therefore identify NGS reads that originate from phases the target read does not stem from and remove them from further consideration. In Fig. 1, the blue and the red NGS reads are filtered out because characteristic SNPs (blue and red tics) could be identified. Mismatches affecting the yellow NGS reads (stars) all lack coverage support, so yellow reads stay in the alignment pile.

In *step 3*, the now cleaned and "phase-unique" set of short reads is divided into small segments the coordinates of which are given by the overlaps of the short reads with the target long read, which acts as a virtual coordinate system. Each of the segments gives rise to a window like part set of NGS reads that overlap the target long read; the segment of the target read itself in a particular window is referred to as "target subread" in the following.

Subsequently, in *step 4,* we now virtually discard the target read for the time being. For each window, we then compute a partial order alignment (POA) [38] of the NGS read segments that pertain to the particular window. Note that the segmentation into window-based target subreads is crucial, because a POA, which is a particular, graph-based type of multiple alignment, tends to accumulate artifacts if not confined to small genomic regions, as is usual for multiple alignments that span too long sequences. The POAs of the short segment subreads then are the foundation for a (phase-unique!) consensus of the short read segments that make part of the POA. Note that computing such a consensus virtually reflects to construct an OG from the POA graph (which is an immediate procedure, as the POA graph spells out all necessary overlaps) and traverse that OG for assembling the short read segments into the desired consensus sequence. These consensus sequence segments establish the corrected sequence of the target subread; therefore, one replaces the original target subread with that consensus segment, to obtain an error corrected version of the target subread.

Finally, in *step 5*, the corrected "target subreads" are re-extended into the complete, full-length reads from which they were computed. Because the target subreads have now been successfully corrected, the full-length read is corrected as well.

*Step 6* then consists of collecting all corrected target reads. In our experiments, we found that iterating the MA/OG stage three times lead to optimal results. This explains why we refer to HERO as the protocol that runs the MA/OG-based sequence of steps 1–6 three times. The resulting corrected TGS reads are the output of HERO.

Note, finally, that *one can iterate this process in an overarching manner* and re-consider the resulting corrected reads as input to the correction procedure (which consists of 3 MA iterations in itself) another time. In that, one can now vary whether one makes repeated use of the DBG stage, the MA stage, or both of them. We make use of such ideas in the following as well. As a general insight, we found that repeated application of the DBG stage (for refined, optimized removal of indel artifacts) followed by repeated application of the MA stage (for refined resolution of phases) had the potential to yield further improvements.

### Data and experimental setup

The datasets that we deal with can be classified into three parts.

The first part refers to simulated metagenomic data, which includes 6 spiked-in datasets. As described in full detail in the "Methods" section, depending on numbers of strains included, we distinguish between simulated datasets of "3 *Salmonella* strains," "20 bacterial strains," and "100 bacterial strains." In addition, we consider the just mentioned "strain-mixing spike-in" datasets, which result from spiking real data with simulated reads generated from known *Salmonella* strains.

The second part refers to the real metagenomic datasets. To evaluate the performance of HERO and prior state-of-the-art approaches in real data scenarios, we considered the two datasets Bmock12 and NWCs (for details, see the "Methods" section), because they have available reference genomes (so available ground truth), and all of Illumina, PacBio CLR, and ONT reads ready for download.

The third part refers to real diploid/polyploid plant genome datasets. In detail, we considered the diploids *Arabidopsis thaliana*, *Arctia plantaginis*, *Oryza sativa japonica*, and *Oryza sativa aromatic*, as well as the tetraploid mango (*Mangifera indica*).

Note that the lack of applicable haplotype-resolved reference genomes for the plant genomes requires to evaluate the experiments in a different way: instead of MetaQUAST, we apply Merqury [39] for evaluating results, as an approved tool for such scenarios.

We refer the reader to the "Methods" section for technical details on the datasets.

### Experiments: simulated metagenomic data

In the following, headings of paragraphs refer to different datasets. For detailed descriptions of the datasets, see the "Methods" section. In general, experiments are discussed in the order by which datasets are explained in the "Methods" section.

### *3 Salmonella strains*

This dataset consists of simulated reads generated from 3 *Salmonella* strains; see the "Methods" for details (Fig. 2). The dataset reflects a simple scenario that allows us to have a clear view on basic adjustments of our workflow. Following our workflow (see Fig. 1), we first pre-corrected the TGS reads using one of the DBG-based HEC methods LoRDEC [13], FMLRC [14], and Ratatosk [15]. Fig. 3a, c display the error content of the TGS reads that remains after iterative application of one of the three DBG-based HEC methods, where the maximum number of iterations was 8. All FMLRC, LoRDEC, and Ratatosk have in common that results markedly improve during the first 3 iterations, and stabilize thereafter; no significant further improvements can be observed after the third iteration. The only exception to that rule is Ratatosk, which exhibited further, albeit only slight improvements beyond the third iteration with respect to indel error content; note however as well that Ratatosk was outperformed by the other two methods with respect to indel error content, opposite to results achieved on mismatch content, where Ratatosk outperforms the other two tools. Based on monitoring performance relative to the number of iterations, and in comparison of the DBG-based tools with each other, we determined to make use of HERO's MA/OG stage after exactly 3 DBG-based iterations, regardless of which DBG-based method was employed. Based on the results achieved here, we suggest to make use of 3 iterations of a DBG-based HEC method prior to applying the MA/OG stage of HERO in general, and stick to that scheme of iterations in the following.

As shown in Fig. 3b, we make use of the MA/OG stage of HERO only after threefold application of DBG-based HEC tools. In that, we iterate application of the MA/OG stage of HERO as well and monitor the resulting performance rates in terms of indel and mismatch content of TGS reads. As one can see in Fig. 3b, the indel error content drops rather dramatically already after the first iteration of the OG stage of HERO (note that one iteration, by definition, consists of 3 repetitions of steps 1–6 of the

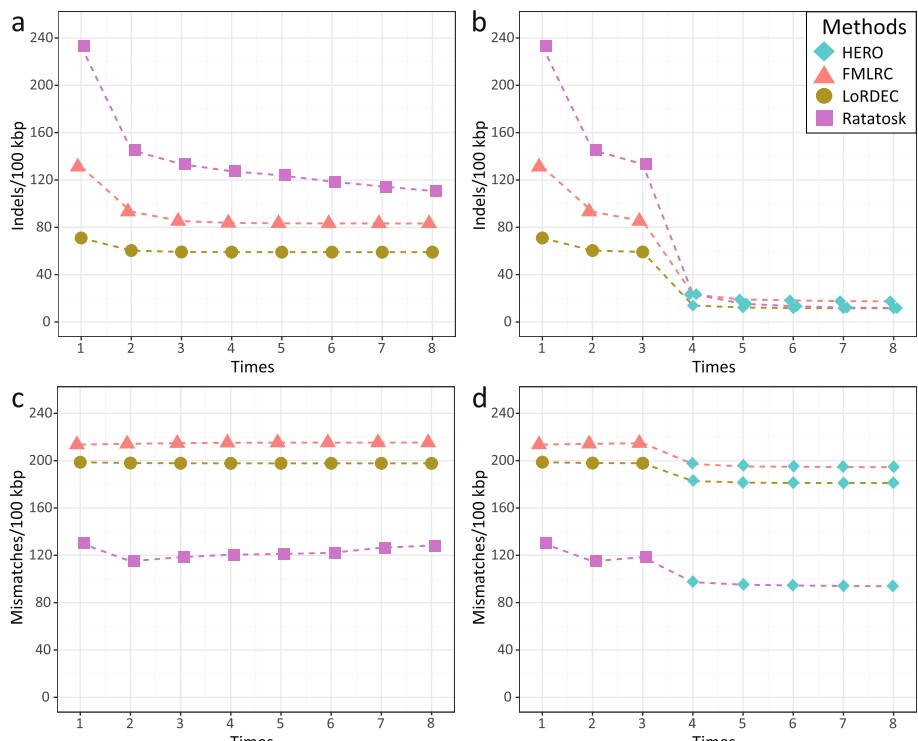

**Fig. 2** Experiments to determine appropriate amounts of iterations for both DBG-based pre-correction of long reads, as performed by FMLRC, LoRDEC, and Ratatosk, and subsequent OG-based correction by HERO-OG, as per steps 1 to 6 in Fig. 1 . The x -axis indicates the different iterations. The y -axis indicates the resulting error rates. Experiments were performed on dataset containing simulated PacBio CLR reads from 3 Salmonella strains. In **a** and **c**, only FMLRC, LoRDEC, and Ratatosk were used. In **b** and **d** , 3 iterations of FMLRC, LoRDEC and Ratatosk are followed by 5 iterations of HERO-OG. Results point out that for DBG-only-based approaches, 8 iterations are optimal, while for hybrid (DBG and OG) approaches, 3 iterations of DBG, followed by 5 iterations of OG yield optimal error rates

workflow of HERO; see Fig. 1). Beyond the first iteration of HERO's OG stage, results keep improving, but only to minor (likely negligible) amounts, regardless of which DBG-based tool was used beforehand.

See Fig. 3d: when considering mismatches, one also observes a marked loss already after the first iteration of HERO's OG stage. Beyond the first iteration, further improvements appear to be negligible, in any case minor with respect to the improvements that one achieves at the beginning.

Given that indels are the predominant type of errors that affect TGS reads and given that HERO induces a fairly drastic reduction of the indel error content of TGS reads, HERO arguably makes a substantial contribution to HEC of TGS reads.

As supported by these basic experiments, we henceforth decided to run 3 DBG-based pre-correction (indel removal) cycles, followed by 5 OG-based correction (haplotype-aware identification of errors). To juxtapose usage of MAs/OGs with using DBGs alone, we sometimes run 5 further DBG cycles instead of 5 MA/OG cycles, because the equal amount of iterations makes for an optimally fair comparison of alternative protocols.

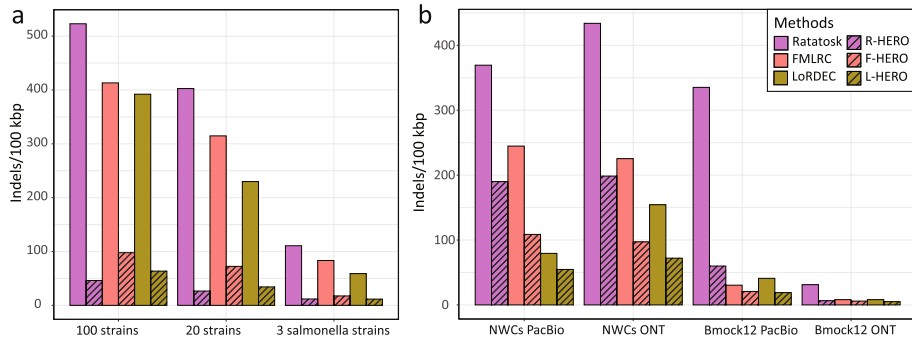

**Fig. 3** Indel error rates (y -axis) for PacBio CLR and ONT reads after correction with different protocols for different datasets (x -axis). Ratatosk, FMLRC, and LoRDEC refer to 8 iterations of the respective methods (pointed out as optimal protocol earlier); R-HERO, F-HERO, and L-HERO refer to 3 iterations of Ratatosk, FMLRC, and LoRDEC, respectively, followed by 5 iterations of HERO-OG. **a** The simulated PacBio CLR reads. **b** Both PacBio CLR and ONT reads from the 4 real datasets

For further, compact summaries of results on the basic 3 *Salmonella* data set, see also the right set of bars in Fig. 4 in the uppermost block of rows in Table 1. One can see that application of 3 times LoRDEC as DBG-based pre-correction stage followed by 5 times the MA/OG stage of HERO leads to a reduction from 6053 mistaken gap positions in the PacBio CLR reads to only 11.58 mistaken gap position per 100 kbp, that is, HERO as per its optimal protocol reduces indel errors by a factor of approximately 600. Note that applying LoRDEC, Ratatosk, or FMLRC alone leave one with 5.5 times more indels as the best result (8 iterations of LoRDEC). The advantages of HERO over DBG-only pro-tocols are obvious across all the different state-of-the-art approaches LoRDEC, Ratatosk, and FMLRC. HERO has ten times better indel error rates over Ratatosk alone (11.77 ver-sus 110.75 per 100 kbp) and 5 times better indel error rates when applying FMLRC alone (17.47 versus 83.26); for LoRDEC, see above (5.5 times better error rates).

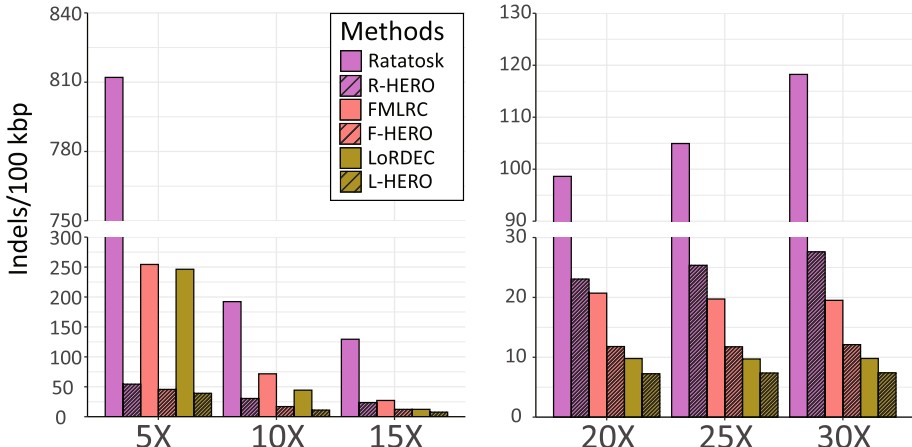

**Fig. 4** Indel error rates (y -axis) versus coverage (x -axis) for the different method protocols. Ratatosk, FMLRC, and LoRDEC refer to 8 iterations of the respective methods (pointed out as optimal protocol earlier); R-HERO, F-HERO, and L-HERO refer to 3 iterations of Ratatosk, FMLRC, and LoRDEC, respectively, followed by 5 iterations of HERO-OG (again pointed out as optimal protocol earlier). The figure shows that the advantages of HERO become more striking on decreasing coverage

**Table 1** Benchmark results for correcting simulated PacBio CLR reads. Indels/100 kbp: average number of insertion or deletion errors per 100,000 aligned bases. Mismatches/100 kbp = average number of mismatch errors per 100,000 aligned bases. Genome fraction (GF) reflects how much of each of the strain-specific genomes is covered by the corrected reads. N/100 kbp denotes the average number of uncalled bases (Ns) per 100,000 bases in the read. MC = fraction of misassembled contigs

| Correction | Indels/100 kbp | Mismatches/100 kbp | GF (%) | N/100 kbp | MC (%) |
|---|---|---|---|---|---|
| 3 *Salmonella* strains | | | | | |
| PacBio CLR reads (10X) | 6053.09 | 1564.71 | 99.96 | 0.00 | 0.00 |
| Ratatosk | 110.75 | 128.23 | 99.96 | 6.53 | 0.29 |
| R-HERO | 11.77 | **93.90** | 99.96 | 0.76 | 0.20 |
| FMLRC | 83.26 | 215.29 | 99.96 | 0.00 | 0.11 |
| F-HERO | 17.47 | 194.45 | 99.96 | 0.00 | 0.13 |
| LoRDEC | 59.00 | 197.74 | 99.95 | 0.00 | 0.00 |
| L-HERO | **11.58** | 181.06 | 99.96 | 0.00 | 0.01 |
| 20 bacteria strains | | | | | |
| PacBio CLR reads (10X) | 6123.04 | 1594.90 | 99.32 | 0.14 | 0.00 |
| Ratatosk | 402.71 | 171.31 | 99.02 | 3.34 | 0.06 |
| R-HERO | **26.54** | **103.51** | 98.47 | 0.43 | 0.13 |
| FMLRC | 314.82 | 325.05 | 96.09 | 0.02 | 0.99 |
| F-HERO | 72.40 | 275.34 | 95.46 | 0.00 | 1.11 |
| LoRDEC | 229.92 | 267.60 | 97.15 | 0.00 | 0.015 |
| L-HERO | 34.25 | 230.43 | 96.51 | 0.00 | 0.182 |
| 100 bacteria strains | | | | | |
| PacBio CLR reads (10X) | 6194.76 | 1644.52 | 98.07 | 0.00 | 0.001 |
| Ratatosk | 523.07 | 267.42 | 99.45 | 6.30 | 0.29 |
| R-HERO | **46.10** | **250.16** | 98.42 | 1.04 | 0.85 |
| FMLRC | 413.23 | 561.24 | 94.27 | 0.00 | 2.31 |
| F-HERO | 97.84 | 511.03 | 94.79 | 0.00 | 3.47 |
| LoRDEC | 392.21 | 517.00 | 96.36 | 0.00 | 0.14 |
| L-HERO | 63.56 | 470.06 | 96.07 | 0.00 | 1.10 |

Boldface is meant to indicate the best performing tool in the respective category

Results for mismatch rates are not as striking as for indel error rates. R-HERO yields 93.9/100 kbp mismatch errors, an improvement of more than 25% over the Ratatosk only (128.23 per 100 kbp). All other conceivable protocols leave at least 180 per 100 kbp mismatches in the TGS reads. R-HERO, F-HERO, and L-HERO refer to 3 iterations of Ratatosk, FMLRC, and LoRDEC, respectively, followed by 5 iterations of HERO-OG.

### *20 bacterial strains*

This dataset consists of 20 bacterial strains, stemming from 10 species (so on average each species has two strains), at an average coverage of 10X per strain for both Illumina (NGS) and PacBio (TGS) reads; see the "Methods" for details. The dataset is supposed to provide a scenario of medium difficulty. See again Fig. 4 and Table 1. Application of R-HERO improves the indel error rate (26.54) by 9 times over the best DBG-only protocol (LoRDEC: 229.92), which again is a rather drastic improvement. As for mismatches, R-HERO improves the mismatch error rate (= 103.51) by 40% with respect to the best DBG-only protocol (Ratatosk: 171.31), which is still quite remarkable.

### 100 bacterial strains

This dataset consists of 100 strains from 30 species, at an average coverage of 10X per strain for both NGS (Illumina) and TGS (PacBio CLR) reads. The dataset is supposed to reflect a more complex scenario. The idea is to evaluate which of the possible protocols and methods potentially become confused by the complexity of the data. In fact (see again Fig. 4 and Table 1), the error rates indeed increase with respect to the two easier datasets just discussed. However, the optimal indel error rate achieved by R-HERO (46.10) stays approximately 9 times better than the best result achieved by DBG-only-based protocols (LoRDEC: 392.21), corroborating the clear advantages of HERO over the state-of-the-art. As for mismatches, R-HERO achieves only slight advantages (250.16) over prior approaches (Ratatosk: 267.42).

In summary, the advantages of HERO over prior approaches/protocols are most evident for indel error rates, with rates improving by a factor of 5 or even more. Advantages for mismatch rates are still clearly evident but are not as striking as for indels. In this vein, it may be important to remember that artificial indels are the predominant error affecting TGS reads.

As for the influence of numbers of species and strains on the results, HERO preserves its advantages across the different levels of complexity in comparison with prior methods (about 9 times less indels regardless of the level of complexity). However, from an absolute point of view, the error rate increases on increasing complexity: from 11.58 for low via 26.54 for medium to 46.10 for more difficult datasets. In contrast, the error rates increase from 59.0 via 229.92 to 392.21 for the prior DBG-only methods. To complete the picture, we recall that the initial indel error rate of the raw reads was greater than 5000 for all datasets.

### Strain-mixing spike-in datasets: the influence of coverage

For investigating the effect of the sequencing coverage of the NGS reads on error correction, we considered 6 "strain-mixing spike-in" datasets. These datasets are characterized by mixing simulated reads of 10 known *Salmonella* strains with real sequence and then correct the errors in the simulated reads; note that we know the ground truth only for the simulated reads, while the majority of reads is real, which is the idea of such datasets. The 6 datasets varied in terms of the coverage of the "spiked-in" NGS reads, which ranged from 5X to 30X in steps of 5X, across the 6 datasets. The average coverage of PacBio CLR reads for the 10 *Salmonella* in the 6 datasets amounted to 10X.

See Fig. 5 for results with respect to indel and mismatch error rates, and see also Additional file 1: Table S1 for full details. If the coverage is below 15X, the reduction of the indel error rate of HERO is particularly noticeable. Results are particularly striking for a coverage of only 5X of the NGS (Illumina) reads. Here, application of L-HERO yields a 6-fold improvement over the best DBG-only method (LoRDEC): 39.09 by L-HERO versus 246.33 by LoRDEC.

On increasing NGS read coverage, all methods, DBG-only and HERO, improve, and differences become less striking as for low coverage. However, even if the coverage of the NGS reads is large, HERO still induces substantial improvements. At 30X, the

| Strain ID | Coverage | Ratatosk | R-HERO | FMLRC | F-HERO | LoRDEC | L-HERO |
|---|---|---|---|---|---|---|---|
| NWCs ONT indel error rate (per 100 kbp) | | | | | | | |
| CP029252.1 | 56.29X | 151.93 | 92.20 | 67.29 | 26.39 | 43.45 | 29.33 |
| CP031021.1 | 55.07X | 155.09 | 94.53 | 65.34 | 28.06 | 46.75 | 31.88 |
| CP029250.1 | 39.38X | 356.00 | 289.75 | 167.17 | 85.33 | 80.74 | 49.46 |
| CP031023.1 | 35.13X | 393.84 | 273.53 | 203.58 | 110.63 | 72.99 | 46.98 |
| CP031018.1 | 17.59X | 635.73 | 232.07 | 320.06 | 133.64 | 206.07 | 86.26 |
| CP031016.1 | 10.27X | 909.16 | 419.45 | 423.57 | 212.24 | 301.57 | 160.84 |
| NWCs ONT mismatch error rate (per 100 kbp) | | | | | | | |
| CP029252.1 | 56.29X | 214.60 | 182.25 | 93.12 | 68.35 | 207.76 | 198.53 |
| CP031021.1 | 55.07X | 235.67 | 207.31 | 110.59 | 88.46 | 240.77 | 228.11 |
| CP029250.1 | 39.38X | 603.53 | 565.49 | 236.08 | 193.32 | 536.88 | 376.55 |
| CP031023.1 | 35.13X | 539.37 | 517.75 | 273.26 | 219.06 | 391.08 | 276.23 |
| CP031018.1 | 17.59X | 719.79 | 446.61 | 415.75 | 321.84 | 535.67 | 393.37 |
| CP031016.1 | 10.27X | 1096.81 | 797.8 | 713.57 | 609.21 | 846.51 | 672.30 |
| Bmock12 ONT indel error rate (per 100 kbp) | | | | | | | |
| 2615840527 | 618.76X | 31.77 | 2.86 | 1.64 | 1.70 | 4.53 | 2.91 |
| 2623620618 | 579.87X | 23.24 | 6.33 | 6.46 | 4.83 | 7.04 | 4.93 |
| 2623620617 | 507.08X | 20.75 | 6.56 | 6.58 | 5.27 | 6.90 | 5.39 |
| 2615840697 | 447.83X | 17.95 | 4.71 | 4.54 | 3.72 | 5.99 | 4.20 |
| 2617270709 | 425.47X | 32.69 | 5.92 | 3.92 | 3.26 | 10.03 | 6.93 |
| 2615840601 | 170.59X | 46.26 | 15.99 | 6.48 | 3.91 | 5.36 | 3.22 |
| 2616644829 | 135.05X | 21.97 | 6.84 | 6.95 | 5.71 | 8.49 | 5.67 |
| 2615840533 | 78.32X | 24.30 | 3.81 | 11.02 | 9.26 | 6.32 | 3.94 |
| 2615840646 | 31.90X | 45.34 | 7.31 | 31.32 | 25.23 | 17.03 | 10.53 |
| 2623620567 | 18.19X | 145.54 | 16.65 | 73.77 | 47.30 | 33.33 | 9.31 |
| 2623620557 | 14.91X | 174.06 | 13.64 | 136.14 | 67.74 | 75.03 | 12.07 |
| Bmock12 ONT mismatch error rate (per 100 kb) | | | | | | | |
| 2615840527 | 618.76X | 20.55 | 3.89 | 1.80 | 1.87 | 4.06 | 3.25 |
| 2623620618 | 579.87X | 159.27 | 71.46 | 28.37 | 27.77 | 34.18 | 33.17 |
| 2623620617 | 507.08X | 169.18 | 78.82 | 38.35 | 38.86 | 35.76 | 35.09 |
| 2615840697 | 447.83X | 37.05 | 19.50 | 13.58 | 13.96 | 15.94 | 15.83 |
| 2617270709 | 425.47X | 34.48 | 13.73 | 6.69 | 6.52 | 13.54 | 11.42 |
| 2615840601 | 170.59X | 40.78 | 20.71 | 8.48 | 7.34 | 7.74 | 6.70 |
| 2616644829 | 135.05X | 70.71 | 42.86 | 29.42 | 29.23 | 29.66 | 27.97 |
| 2615840533 | 78.32X | 28.37 | 7.30 | 11.95 | 10.66 | 6.35 | 5.12 |
| 2615840646 | 31.90X | 38.79 | 7.60 | 33.66 | 29.03 | 25.12 | 19.81 |
| 2623620567 | 18.19X | 214.45 | 60.19 | 101.34 | 84.93 | 68.70 | 49.33 |
| 2623620557 | 14.91X | 225.95 | 49.62 | 164.45 | 116.45 | 110.04 | 56.50 |

**Fig. 5** Indel error rates for different methods stratified by the strains of the real datasets (1st column: NWC = Genbank ID, Bmock12 = IMG Taxon ID), which are ordered in descending order by their coverages (2nd column). Error rates are colored from large (red) via medium (white) to low (blue). The evident trend are improvements in indel error rate relative to increasing coverage

indel error rate of L-HERO reduces that of LoRDEC by 24.49% (LoRDEC: 9.8 versus L-HERO: 7.4). Note that the mismatch error rate of FMLRC is the best among all DBG-only approaches: 107.8 versus 208.3 by LoRDEC, at a coverage of 30X. Although F-HERO corrected mismatch error is close to FMLRC (F-HERO:102.87 versus FMLRC:107.8), F-HERO reduces indel errors by up to 37.93% over FMLRC only (F-HERO: 12.11 versus FMLRC: 19.51).

### Experiments: real metagenomic datasets

We further evaluated all approaches on the 4 real datasets "Bmock 12 PacBio", "Bmock 12 ONT," "NWC PacBio," and "NWC ONT."

#### *Bmock12 ONT*

The "Bmock12" dataset contains 11 strains from 9 species, which is of rather low complexity. The two species that exhibit more than 1 strain are *Marinobacter* and *Halomonas.*

See Table 2 for the following results. All methods achieve relatively low error rate. As supported by our experiments on the spike-in datasets with respect to coverage, the

**Table 2** Benchmark results for correcting reads from real datasets. Indels/100 kbp: average number of insertion or deletion errors per 100,000 aligned bases. Mismatches/100 kbp = average number of mismatch errors per 100,000 aligned bases. Genome fraction (GF) reflects how much of each of the strain-specific genomes is covered by the corrected reads. N/100 kbp denotes the average number of uncalled bases (Ns) per 100,000 bases in the read. MC = fraction of misassembled contigs

| Correction | Indels/100 kbp | Mismatches/100 kbp | GF (%) | N/100 kbp | MC (%) |
|---|---|---|---|---|---|
| Correct Bmock12 ONT reads | | | | | |
| ONT reads | 4001.58 | 2659.11 | 93.63 | 0.00 | 2.33 |
| Ratatosk | 31.17 | 94.91 | 93.74 | 0.67 | 2.64 |
| R-HERO | 6.45 | 40.48 | 93.78 | 0.20 | 2.57 |
| FMLRC | 8.09 | 21.82 | 93.77 | 0.00 | 2.93 |
| F-HERO | 5.87 | 20.82 | 93.77 | 0.06 | 2.92 |
| LoRDEC | 8.10 | 22.56 | 93.76 | 0.00 | 2.73 |
| L-HERO | **5.03** | **20.54** | 93.81 | 0.06 | 2.70 |
| Correct Bmock12 PacBio reads | | | | | |
| PacBio reads | 7790.78 | 1851.43 | 87.98 | 0.00 | 7.55 |
| Ratatosk | 335.23 | 152.21 | 91.76 | 1.16 | 10.53 |
| R-HERO | 59.87 | 58.89 | 92.31 | 0.25 | 12.28 |
| FMLRC | 30.39 | 46.53 | 94.84 | 0.00 | 18.14 |
| F-HERO | 20.76 | **42.45** | 94.88 | 0.03 | 18.55 |
| LoRDEC | 40.99 | 60.44 | 93.00 | 0.00 | 15.32 |
| L-HERO | **18.95** | 53.01 | 93.51 | 0.04 | 17.09 |
| Correct NWC ONT reads | | | | | |
| ONT reads | 7991.02 | 5476.30 | 100.00 | 0.00 | 7.18 |
| Ratatosk | 433.92 | 589.11 | 99.99 | 22.42 | 7.22 |
| R-HERO | 198.49 | 426.52 | 100.00 | 13.25 | 7.49 |
| FMLRC | 225.39 | 340.97 | 99.99 | 0.00 | 10.35 |
| F-HERO | 97.17 | **270.36** | 99.99 | 0.00 | 10.70 |
| LoRDEC | 154.4 | 489.82 | 99.99 | 0.00 | 7.18 |
| L-HERO | **72.05** | 377.92 | 99.99 | 0.00 | 9.65 |
| Correct NWC PacBio reads | | | | | |
| PacBio reads | 11479.15 | 5270.85 | 81.61 | 0.00 | 7.54 |
| Ratatosk | 369.40 | 446.14 | 83.13 | 13.65 | 6.12 |
| R-HERO | 190.04 | 301.22 | 83.45 | 8.66 | 5.58 |
| FMLRC | 244.74 | 239.31 | 84.62 | 0.00 | 5.14 |
| F-HERO | 108.63 | **171.05** | 84.49 | 0.00 | 5.01 |
| LoRDEC | 79.42 | 422.12 | 83.19 | 0.00 | 5.01 |
| L-HERO | **54.68** | 373.06 | 83.00 | 0.00 | 4.76 |

Boldface is meant to indicate the best performing tool in the respective category

most plausible explanation is the fact that the NGS read coverage was relatively large (15X to 618X), in combination with the fairly low complexity of the dataset. Still, however, application of HERO exhibited substantial improvements over the results that DBG-only-based methods achieved. For example, HERO reduced the indel error rate by on average 48% (27~79%) in direct juxtaposition (e.g., FMLRC versus F-HERO) with the DBG-only protocols. L-HERO outperformed the other approaches in particular——its indel error rate is 37.9% lower than that of LoRDEC only: 5.03 by L-HERO versus 8.10 by LoRDEC. As for mismatch errors, L-HERO reduces errors by only 9% in comparison with LoRDEC: 20.54 by L-HERO versus 22.56 by LoRDEC.

### Bmock12 PacBio

For maximal comparability, we used the same Illumina reads as for ONT to also correct the PacBio CLR reads. We found that the error rate of the uncorrected TGS reads affected the quality of the correction. Note that the indel error rate of the Bmock12 PacBio CLR data is 60% larger than that of the ONT reads (4001.58 by ONT versus 7790.78 by PacBio CLR). Although we used the same Illumina reads to correct the PacBio CLR reads, more indel errors were retained in comparison with the ONT reads. Apart from that difference, results are largely analogous to the results obtained for ONT reads. In particular, after correction with Ratatosk, the PacBio CLR reads still exhibited an indel error rate of 335.23 per 100 kbp, which is 10 times larger than Ratatosk on ONT reads (31.17). As for levels of improvement of HERO over DBG-only methods, results scale similarly in a relative comparison. For example, L-HERO achieves an indel rate of 18.95 versus 40.99 by LoRDEC, so it reduces the errors by 50% over the DBG-only approach. We recall that for ONT reads, HERO achieved a relative improvement of 37.9% (see above: 5.03 versus 8.10).

### NWC

This dataset contains 3 species (*Streptococcus thermophilus*, *Lactobacillus delbrueckii*, *Lactobacillus helveticus*), of 2 strains each, of ANI 99.99%, 99.24%, and 98.03%, respectively. Because of the high ANI, this dataset is particularly challenging in terms of strain diversity; see Additional file 1: Table S3 for additional information. Although we removed the low quality bases (> Q20), the NGS (Illumina) reads retained an unusually high error rate (indel error rate: 19.58/100 kbp; mismatch error rate: 883.28). This does not reflect standard scenarios and can have a considerable impact on the quality of the correction. To re-establish a standard scenario, we corrected the NGS reads prior to hybrid correction of the TGS reads, by using bfc [40], an approved error corrector applicable for Illumina short reads.

### NWC ONT

Here, we discuss the results from using ONT reads as TGS reads, the coverage of which was 94.57X. As one can see in Fig. 4b and Table 2, the correction results for the NWC datasets were worse than for Bmock12 and the simulated datasets. The obvious explanation is the difficulty of this dataset in terms of the similarity of the strains (as expressed by large ANIs, see above). Moreover, the initial error rate of the raw ONT reads was substantially larger than for Bmock12.

Still, HERO reduced the indel error rate by more than 50% (a factor of 2) in direct comparison with the toughest competitor (L-HERO: 72.05; LoRDEC: 154.4), which means a remarkable reduction nevertheless. In particular, this means that the indel error rate drops to less than 1% of that of the raw reads (7991.02). As for mismatch correction, F-HERO outperformed the other methods, reducing it by more than 20% in comparison with the best DBG-only protocol (F-HERO: 270.36; FMLRC: 340.97). It also means that mismatch errors drop to approximately 5% of the original value (5476.30).

### NWC PacBio

See Table 2, L-HERO outperformed all other approaches in terms of indel error correction, and drops the error rate by 31.15% in direct comparison with the best DBG-only approach (L-HERO: 54.68 versus LoRDEC: 79.42). In terms of correcting mismatches, F-HERO outperforms the other approaches. In direct comparison with FMLRC, as the toughest DBG-only approach, F-HERO achieves a reduction of 28.52% (F-HERO: 171.05; FMLRC: 239.31).

### NGS read coverage

To evaluate the effect of NGS (here: Illumina) read coverage, we analyzed how performance in the 4 real datasets varied relative to variations thereof; see Fig. 6 and Additional file 1: Fig. S3. Note that the NGS read coverage of the 6 strains in the NWC dataset is not even but ranges from 10.27X to 56.29X. We therefore just stratified correction results to the individual 6 strains and related individual correction rates to the individual coverages of the strains. We applied the same stratification analysis for the 11 strains making part of Bmock12, whose coverages range from 14.91X to 618.76X.

The first and obvious observation is the fact that the error rates correlate negatively with NGS read coverage: the larger the coverage and the lower the error rate. The second observation is that the general trends with respect to differences between the datasets Bmock12 and NWC persist in this analysis: results are generally better for Bmock12, owing to the reduced level of difficulty in terms of the similarity of the strains in Bmock12.

As for the negative correlation between error rates and coverage, see Fig. 6, where F-HERO is an exemplary protocol: the lower the coverage, the higher the error rates, as in NWC, where each strain is partnered by another one from the same species. This correlation is disturbed for Bmock12, because the majority of the 11 strains has no counterpart, which renders error correction substantially easier for them; this then further has a clear influence on the error rates. Still, however, as a look at the exemplary F-HERO points, the negative correlation gets established also there (note that correlation also exists for R-HERO and L-HERO, albeit with some fluctuations).

As for examples of relations of coverage and error rates, note that in the Bmock12 dataset, the indel error rate drops substantially as soon as coverage rates exceed 30X for the NGS reads. This effect can be observed across both datasets, and all methods, both HERO and DBG-only type.

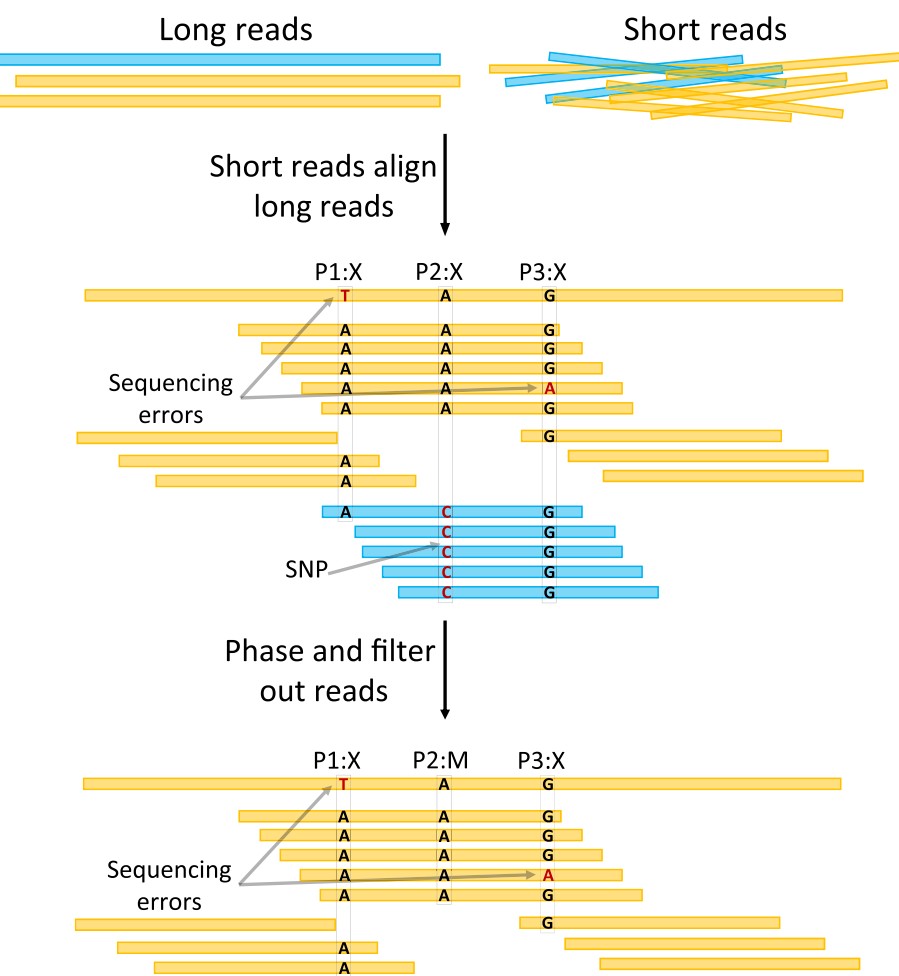

**Fig. 6** Steps 1 and 2 of the workflow displayed in Fig. 1. Step 1, "Short reads align long reads," leads to alignments of the long read to be corrected with short reads. Given the alignments, putative variant positions P1:X, P2:X, and P3:X are identified ("X" indicates mismatch). Step 2, "Phase and filter out reads," inspects the sequence content of both the long and the short reads at each of these positions. Different scenarios are conceivable. In P1:X, all short reads agree on "A," so no short reads are removed because of this position. In P2:X, there are large amounts of short reads for supporting both "A," agreeing with the long read, and "C," disagreeing with the long read. Because the number of short reads in disagreement with the long read is too large, these short reads are removed ("filtered out"); most likely, they stem from a different haplotype. We now refer to this position as P2:M with "M" meaning "match," indicating that there is nothing further to be done. In P3:X, all but one short read support "G." Because the amount of short reads (exactly one) that do not support "G" is too small, these short reads are kept; most likely, they contain a sequencing error. After these two steps, P1:X and P3:X require further attention, while P2:M has been resolved

## Improving genome assembly

The arguably canonical area of application that decisively depends on sequencing reads that are both long and sufficiently free of errors is de novo genome assembly. It also stands out insofar as reads are its only input. We therefore pay particular attention to the effects of our hybrid-hybrid error correction framework on the assemblies that one obtains. For maximum comparability, we concentrated on Canu [20] in particular as an assembly tool that (1) reflects the state-of-the-art as a very popular current tool and (2) offers an option to compute assemblies from PacBio HiFi reads ("HiCanu") [41]. As for (2), our results demonstrated that our corrected TGS reads

are substantially lower than the error rates of PacBio HiFi reads—on a side remark, note that this establishes a major advantage over PacBio HiFi reads themselves, as the length of PacBio CLR and ONT reads is on average two and four times greater, respectively. This justifies to run HiCanu on our corrected reads. Subsequently, one can compare the assemblies using raw reads using Canu itself (which integrates an error correction module in its own right) with the assemblies using the corrected reads using HiCanu. Furthermore, we considered Hifiasm-meta [24], which cannot process the uncorrected reads, which does not allow a comparison on the improvement of assemblies before and after correction of errors. We considered Hifiasm-meta mostly to demonstrate that the improvements we claim to make do not depend on a particular assembler.

To highlight the particular advantages of our hybrid-hybrid error correction protocol, we further focused on scenarios that exhibit multiple, closely related genomes. Therefore, we considered the 4 real datasets that we considered earlier (Bmock12-PacBio, Bmock12-ONT, NWC-PacBio, NWC-ONT).

See Table 3 for results on Canu/HiCanu and Table 4 for results on Hifiasm-meta. The first observation is that HERO induces considerable improvements of the assemblies over assemblies based on DBG-only hybrid error corrected reads, both in terms of genome fraction and error content. When running HiCanu, HERO also yields improvements in terms of the length of the contigs, summarized here by the NGA50. When running Hifiasm-meta, there are no improvements in terms of sheer length (while the considerable improvements in terms of genome fraction and error content persist, as above-mentioned).

For example, on the Bmock12 ONT dataset, the performance of HiCanu using R-HERO corrected reads leads to increases of 6.79% in terms of genome fraction with respect to Canu itself and to 2.36% over the corresponding DBG-only approach (R-HERO + HiCanu: 63.92% versus Ratatosk + HiCanu: 61.56% versus Canu: 57.13%). The relative indel error rate is reduced by 400 times over Canu and more than halves that of the corresponding DBG-only approach (R-HERO: 1.27 versus Ratatosk: 2.99 versus Canu: 504.95). Moreover, R-HERO more than halves the mismatch error rate of Ratatosk (34.3 versus 83.12; note however that F-HERO achieves a mismatch error rate of 6.9, obviously empowered by the error rate of 8.76 achieved by FMLRC itself). Last but not least, NGA50 of R-HERO + HiCanu (198551) is greater than that of Ratatosk + HiCanu (137201), and the misassembly rate of R-HERO + HiCanu (2.7) is lower than that of Ratatosk + HiCanu (4.04).

Note that these trends largely persist across the three different DBG-only-based HEC approaches (Ratatosk, FMLRC, LoRDEC), and the different assemblers (Canu/HiCanu, Hifiasm-meta). Rare exceptions, where the error rate slightly increases, or the Genome Fraction slightly drops, are nevertheless possible. Apparently, there is only one more systematic disadvantage of our hybrid-hybrid error correction protocol: with only rare exceptions, the number of uncalled bases (Ns) in the assemblies slightly increases across the scenarios.

**Table 3** Results on assembly after correction by different correction protocols of reads from 4 different real datasets. 1st column: genome assembly program; Canu was used for raw reads, while HiCanu was used for corrected reads (because their error profiles resemble those of PacBio HiFI reads). 2nd column: Error correction protocol. Indels/100 kbp: average number of insertion or deletion errors per 100,000 aligned bases. Mismatches/100 kbp = average number of mismatch errors per 100,000 aligned bases. Genome fraction (GF) reflects how much of each of the strain-specific genomes is covered by the corrected reads. NGA50 is the length of the longest contig such that the alignments of that and all longer contigs span at least 50% of the reference sequence. N/100 kbp denotes the average number of uncalled bases (Ns) per 100,000 bases in the read. MC = fraction of misassembled contigs

| Assemblers | | GF (%) | Indels/100 kbp | Mismatches/100 kbp | NGA50 | N/100 kbp | MC (%) |
|---|---|---|---|---|---|---|---|
| | Bmock12 ONT | | | | | | |
| Canu | ONT raw reads | 57.13 | 504.95 | 140.75 | 78498 | 0.00 | 9.04 |
| HiCanu | Ratatosk | 61.56 | 2.99 | 83.12 | 137201 | 0.22 | 4.04 |
| HiCanu | R-HERO | **63.92** | 1.27 | 34.30 | **198551** | 0.07 | 2.70 |
| HiCanu | FMLRC | 60.64 | 1.10 | 8.76 | 136705 | 0.00 | 4.48 |
| HiCanu | F-HERO | 60.84 | **0.82** | **6.90** | 148570 | 0.03 | 4.23 |
| HiCanu | LoRDEC | 56.48 | 1.43 | 9.92 | 90070 | 0.00 | 4.28 |
| HiCanu | L-HERO | 60.50 | 1.14 | 16.48 | 125982 | 0.03 | 4.74 |
| | Bmock12 PacBio | | | | | | |
| Canu | PacBio raw reads | 50.01 | 174.71 | 63.79 | 6025 | 0.00 | 3.72 |
| HiCanu | Ratatosk | 59.25 | 3.88 | 66.22 | 31170 | 0.46 | 2.61 |
| HiCanu | R-HERO | 61.48 | 1.85 | 38.50 | 42481 | 0.06 | 4.60 |
| HiCanu | FMLRC | 62.54 | 2.02 | 19.50 | **65939** | 0.00 | 2.40 |
| HiCanu | F-HERO | **62.83** | 1.77 | **19.28** | 61786 | 0.03 | 2.45 |
| HiCanu | LoRDEC | 55.47 | **1.67** | 23.59 | 23307 | 0.00 | 2.96 |
| HiCanu | L-HERO | 60.27 | 1.72 | 35.59 | 44240 | 0.01 | 6.51 |
| | NWC ONT | | | | | | |
| Canu | ONT raw reads | - | - | - | - | - | - |
| HiCanu | Ratatosk | 75.74 | 88.44 | 173.91 | 47297 | 5.39 | 42.06 |
| HiCanu | R-HERO | 78.72 | 48.09 | 179.33 | 52662 | 3.05 | 53.51 |
| HiCanu | FMLRC | 84.95 | 66.68 | 180.80 | 61515 | 0.00 | 50.25 |
| HiCanu | F-HERO | **89.12** | 37.92 | 160.21 | **84587** | 0.00 | 50.48 |
| HiCanu | LoRDEC | 36.71 | 42.98 | 185.76 | - | 0.00 | 21.54 |
| HiCanu | L-HERO | 61.66 | **27.54** | **141.29** | 37960 | 0.00 | 31.11 |
| | NWC PacBio | | | | | | |
| Canu | PacBio raw reads | 44.27 | 62.25 | 33.51 | - | 0.00 | 36.51 |
| HiCanu | Ratatosk | 49.66 | 18.72 | 59.16 | 28052 | 2.15 | 6.32 |
| HiCanu | R-HERO | 53.38 | 11.10 | 57.32 | 29365 | 1.97 | 9.52 |
| HiCanu | FMLRC | 51.28 | 20.84 | **32.52** | 29528 | 0.00 | 6.12 |
| HiCanu | F-HERO | **53.39** | 15.98 | 44.49 | **33130** | 0.00 | 11.10 |
| HiCanu | LoRDEC | 50.34 | 8.27 | 56.65 | 26773 | 0.00 | 5.81 |
| HiCanu | L-HERO | 51.04 | **6.43** | 54.91 | 27546 | 0.00 | 5.98 |

Boldface is meant to indicate the best performing tool in the respective category

**Table 4** Results on assembly using Hifiasm-meta after correction by different correction protocols of PacBio CLR and ONT reads from 4 different real datasets. 1st column: genome assembly program = Hifiasm-meta. 2nd column: Error correction protocol. Ratatosk, FMLRC, and LoRDEC refer to 8 iterations of the respective methods (pointed out as optimal protocol earlier); R-HERO, F-HERO, and L-HERO refer to 3 iterations of Ratatosk, FMLRC, and LoRDEC, respectively, followed by 5 iterations of HERO-OG. Indels/100 kbp: average number of insertion or deletion errors per 100,000 aligned bases. Mismatches/100 kbp = average number of mismatch errors per 100,000 aligned bases. Genome fraction (GF) reflects how much of each of the strain-specific genomes is covered by the corrected reads. N/100 kbp denotes the average number of uncalled bases (Ns) per 100,000 bases in the read. MC = fraction of misassembled contigs

| Assemblers | | GF(%) | Indels/100 kbp | Mismatches/100 kbp | NGA50 | N/100 kbp | MC (%) |
|---|---|---|---|---|---|---|---|
| | Bmock12 ONT | | | | | | |
| Hifiasm-meta | Ratatosk | 72.24 | 13.52 | 80.91 | 124596 | 0.19 | **5.85** |
| Hifiasm-meta | R-HERO | **75.81** | **3.12** | 38.09 | 122516 | 0.09 | 6.16 |
| Hifiasm-meta | MLRC | 73.40 | 5.65 | 24.14 | **143247** | 0.00 | 7.84 |
| Hifiasm-meta | F-HERO | 72.24 | 3.96 | **20.79** | 132107 | 0.02 | 7.84 |
| Hifiasm-meta | LoRDEC | 70.17 | 7.14 | 24.42 | 115314 | 0.00 | 7.59 |
| Hifiasm-meta | L-HERO | 71.34 | 3.20 | 21.05 | 118102 | 0.03 | 6.24 |
| | Bmock12 PacBio | | | | | | |
| Hifiasm-meta | Ratatosk | 58.76 | 28.69 | 56.05 | 25507 | 0.36 | 5.40 |
| Hifiasm-meta | R-HERO | 61.97 | 7.80 | 35.78 | 28983 | 0.05 | **4.80** |
| Hifiasm-meta | FMLRC | 67.90 | 5.84 | 20.34 | **33664** | 0.00 | 5.20 |
| Hifiasm-meta | F-HERO | **67.94** | 4.53 | **17.71** | 31602 | 0.01 | 5.60 |
| Hifiasm-meta | LoRDEC | 56.53 | 7.08 | 22.48 | 29667 | 0.00 | 6.02 |
| Hifiasm-meta | L-HERO | 62.42 | **3.56** | 21.58 | 30746 | 0.02 | 5.10 |
| | NWC ONT | | | | | | |
| Hifiasm-meta | Ratatosk | 66.86 | 155.25 | 264.14 | 35297 | 9.25 | 31.69 |
| Hifiasm-meta | R-HERO | **73.99** | 84.48 | 270.01 | 39254 | 5.68 | 29.93 |
| Hifiasm-meta | FMLRC | 70.68 | 52.21 | **105.07** | 46395 | 0.00 | 30.74 |
| Hifiasm-meta | F-HERO | 73.17 | **32.38** | 128.81 | **51599** | 0.00 | **29.17** |
| Hifiasm-meta | LoRDEC | 32.56 | 86.05 | 402.98 | - | 0.00 | 35.63 |
| Hifiasm-meta | L-HERO | 50.23 | 43.10 | 258.81 | 6813 | 0.00 | 42.26 |
| | NWC PacBio | | | | | | |
| Hifiasm-meta | Ratatosk | 54.19 | 141.16 | 450.63 | 25309 | 1.83 | 8.12 |
| Hifiasm-meta | R-HERO | 58.39 | 44.81 | **136.94** | 27327 | 1.01 | **5.03** |
| Hifiasm-meta | FMLRC | 55.88 | 46.17 | 153.2 | 28732 | 0.00 | 6.86 |
| Hifiasm-meta | F-HERO | **59.54** | **31.15** | 180.53 | **29612** | 0.00 | 6.26 |
| Hifiasm-meta | LoRDEC | 38.38 | 48.11 | 327.45 | 9535 | 0.00 | 16.13 |
| Hifiasm-meta | L-HERO | 45.94 | 41.72 | 588.42 | 19398 | 0.00 | 10.55 |

Boldface is meant to indicate the best performing tool in the respective category

## Experiments: real diploid/polyploid genome datasets

Here, we used HERO to correct errors in reads from four diploid genomes (*Arabidopsis thaliana*, *Arctia plantaginis*, *Oryza sativa japonica*, and *Oryza sativa aromatic*) and a tetraploid mango (*Mangifera indica*) genome. See Table 5 for corresponding results.

As above-mentioned, we utilized the reference-free evaluation method Merqury [39] to assess the differences in error rates after correction using different strategies. As you can see in Table 5, applying HERO (F-HERO, R-HERO, and L-HERO) significantly reduces the error rates in comparison with running the DBG-based tools alone. With HERO, error rates were reduced by on average 48.58% (23.44 ~ 71.59%) compared to

**Table 5** Evaluating the effects of different correcting strategies on diploid/polyploid genome sequencing data. In the completeness (%) multicolumn, all is the k-mer completeness for both haplotypes combined, matis maternal hap-mer completeness and pat is paternal hap-mer completeness. Switch error rate = the ratio of the number of switches between true and alternative haplotypes to the total number of variant positions

| Datasets | QV | Error rate (%) | Completeness (%) | | | Switch error rate (%) |
|---|---|---|---|---|---|---|
| | | | Al | Mat | Pat | |
| *Arabidopsis thaliana* | | | | | | |
| PacBio reads | 10.56 | 8.78 | 89.54 | 80.05 | 79.89 | 47.80 |
| Ratatosk | 13.83 | 4.14 | 99.04 | 97.89 | 97.84 | 15.38 |
| R-HERO | 16.04 | 2.49 | 99.08 | 98.01 | 97.94 | **12.23** |
| FMLRC | 15.07 | 3.11 | 97.10 | 92.33 | 92.05 | 17.81 |
| F-HERO | 16.23 | **2.38** | 97.96 | 95.28 | 95.11 | 15.91 |
| LoRDEC | 14.58 | 3.48 | 97.80 | 93.89 | 93.65 | 22.20 |
| L-HERO | 15.94 | 2.55 | 98.43 | 95.94 | 95.76 | 18.70 |
| *Arctia plantaginis* | | | | | | |
| PacBio reads | 10.46 | 9.00 | 97.76 | 37.26 | 36.7 | 57.12 |
| Ratatosk | 14.03 | 3.95 | 98.59 | 63.19 | 62.28 | 55.38 |
| R-HERO | 19.49 | 1.12 | 99.67 | 68.02 | 67.02 | 54.14 |
| FMLRC | 15.08 | 3.10 | 98.90 | 56.36 | 55.52 | 55.88 |
| F-HERO | 19.27 | 1.18 | 99.71 | 65.92 | 65.03 | 54.19 |
| LoRDEC | 15.86 | 2.60 | 98.86 | 58.89 | 58.00 | 54.73 |
| L-HERO | 20.83 | **0.83** | 99.76 | 65.31 | 64.42 | **53.71** |
| *Oryza sativa japonica* | | | | | | |
| ONT reads | 13.47 | 4.50 | 91.14 | — | — | — |
| Ratatosk | 19.41 | 1.14 | 98.37 | — | — | — |
| R-HERO | 21.74 | 0.67 | 99.57 | — | — | — |
| FMLRC | 20.85 | 0.82 | 98.26 | — | — | — |
| F-HERO | 23.79 | 0.42 | 99.32 | — | — | — |
| LoRDEC | 22.10 | 0.62 | 98.65 | — | — | — |
| L-HERO | 25.70 | **0.27** | 99.41 | — | — | — |
| *Oryza sativa aromatic* | | | | | | |
| PacBio reads | 10.30 | 9.34 | 56.29 | — | — | — |
| Ratatosk | 16.79 | 2.10 | 95.84 | — | — | — |
| R-HERO | 19.83 | 1.04 | 99.37 | — | — | — |
| FMLRC | 17.94 | 1.61 | 92.56 | — | — | — |
| F-HERO | 21.99 | 0.63 | 98.11 | — | — | — |
| LoRDEC | 19.35 | 1.16 | 95.50 | — | — | — |
| L-HERO | 23.19 | **0.48** | 98.80 | — | — | — |
| *Mangifera indica* | | | | | | |
| PacBio reads | 10.40 | 9.11 | 96.78 | — | — | — |
| Ratatosk | 17.99 | 1.59 | 99.94 | — | — | — |
| R-HERO | 20.30 | 0.93 | 99.98 | — | — | — |
| FMLRC | 19.85 | 1.04 | 99.86 | — | — | — |
| F-HERO | 21.41 | **0.72** | 99.94 | — | — | — |
| LoRDEC | 18.73 | 1.34 | 99.95 | — | — | — |
| L-HERO | 21.34 | 0.73 | 99.98 | — | — | — |

Boldface is meant to indicate the best performing tool in the respective category

applying the DBG-based iterations only. Especially for *Arctia plantaginis, Oryza sativa japonica, Oryza sativa aromatic*, and *Mangifera indica*, the sequencing error rate even dropped to below 1%, which translates into error rates that commonly apply for NGS reads.

*Arabidopsis thaliana* and *Arctia plantaginis* reflect trio data, so Merqury can establish a database, and by making use of the sequencing data of the parents, one can evaluate the switch error rates. Note that switch errors are an indication of misassemblies. The results that HERO reduces switch error rate.

We proceeded to assemble the reads using Canu and analyzed the impact of the error correction methods/protocols on the resulting genome assemblies; see Table 6 for corresponding results. Although the error rates significantly dropped after correction, the resulting error rates were too large to apply HiCanu for assembly, so we resorted

**Table 6** Assembling the corrected reads using Canu. Assess the influence of different correction methods for diploid/polyploid genome assembly. In the completeness (%) multicolumn, all is the k-mer completeness for both haplotypes combined, matis maternal hap-mer completeness and pat is paternal hap-mer completeness. Switch error rate = the ratio of the number of switches between true and alternative haplotypes to the total number of variant positions

| Assembler | Data | QV | Error rate (%) | Completeness (%) | | | Switch error rate (%) |
|---|---|---|---|---|---|---|---|
| | | | | **All** | **Mat** | **Pat** | |
| *Arabidopsis thaliana* | | | | | | | |
| Canu | Ratatosk | 22.87 | 0.52 | 66.55 | 40.31 | 40.31 | 8.59 |
| Canu | R-HERO | 23.86 | 0.41 | **70.13** | 43.05 | 43.05 | **7.94** |
| Canu | FMLRC | 23.63 | 0.43 | 61.46 | 32.51 | 32.29 | 11.34 |
| Canu | F-HERO | 23.88 | 0.41 | 61.89 | 33.01 | 32.10 | 11.17 |
| Canu | LoRDEC | 25.15 | 0.31 | 48.22 | 25.64 | 26.15 | 12.18 |
| Canu | L-HERO | 25.61 | **0.27** | 58.23 | 32.35 | 32.38 | 11.05 |
| *Oryza sativa japonica* | | | | | | | |
| Canu | Ratatosk | 25.23 | 0.30 | 50.36 | — | — | — |
| Canu | R-HERO | 28.40 | 0.14 | **58.87** | — | — | — |
| Canu | FMLRC | 25.75 | 0.27 | 28.36 | — | — | — |
| Canu | F-HERO | 29.53 | 0.11 | 34.33 | — | — | — |
| Canu | LoRDEC | 25.88 | 0.26 | 30.28 | — | — | — |
| Canu | L-HERO | 29.83 | **0.10** | 54.27 | — | — | — |
| *Oryza sativa aromatic* | | | | | | | |
| Canu | Ratatosk | 24.12 | 0.39 | 12.32 | — | — | — |
| Canu | R-HERO | 29.03 | 0.13 | **58.89** | — | — | — |
| Canu | FMLRC | 30.17 | 0.10 | 1.15 | — | — | — |
| Canu | F-HERO | 32.63 | **0.05** | 3.57 | — | — | — |
| Canu | LoRDEC | 25.76 | 0.27 | 4.38 | — | — | — |
| Canu | L-HERO | 29.94 | 0.10 | 41.28 | — | — | — |
| *Mangifera indica* | | | | | | | |
| Canu | Ratatosk | 24.44 | 0.36 | **90.60** | — | — | — |
| Canu | R-HERO | 26.41 | 0.23 | 90.43 | — | — | — |
| Canu | FMLRC | 28.48 | 0.14 | 75.22 | — | — | — |
| Canu | F-HERO | 30.05 | **0.10** | 75.19 | — | — | — |
| Canu | LoRDEC | 24.67 | 0.34 | 61.46 | — | — | — |
| Canu | L-HERO | 26.88 | 0.21 | 70.09 | — | — | — |

Boldface is meant to indicate the best performing tool in the respective category

to Canu. First, we assembled Arabidopsis thaliana: while we were able to successfully assemble the corrected reads, Canu was not able to assemble the original reads and reported insufficient coverage instead. This points out that correcting errors in TGS reds prior to assembly can be crucial just to get rid of scenarios that are flawed by read coverage artifacts.

For the other organisms, Canu could neither assemble the original nor the corrected reads. To amend the issue, we modified Canu's parameters (so deviating from default parameters), by lowering Canu's minimum depth requirement to 3X. As a consequence, we were able to obtain reasonable assemblies for all organisms except Arctia plantaginis. For all organisms, Canu still failed to assemble the original raw, uncorrected reads. This is another indication that sound error correction is a crucial task when aiming to assemble genomes in a haplotype-aware manner.

As one can see in Table 6, in most cases, the completeness of the assembly increased after correcting reads using HERO. This improvement was especially noticeable for the relatively low coverage samples of *Oryza sativa japonica* (11X) and *Oryza sativa aromatic* (21X). Additionally, the assembly results for *Arabidopsis thaliana* clearly demonstrate that HERO supports to drop the switch error rates, as a clear indication that contigs suffer considerably less from misassemblies.

We also assembled the corrected reads using Hifiasm; see Table 7. Although the *Arabidopsis thaliana* datasets enjoys relatively large sequencing depth, the sequencing error rates remain relatively large after correction, in all cases, with or without HERO. This explains why the completeness of the assembly is rather low. Still, however, applying HERO (R-HERO, F-HERO, or L-HERO) in comparison with just applying Ratatosk, FMLRC, and LoRDEC, points out that the additional application of HERO yields substantial improvements in terms of error content. Similar trends show in the other four organisms, especially in the ones whose sequencing data has low depth: *Arctia plantaginis* (18X), *Oryza sativa japonica* (11X), and *Oryza sativa aromatic* (21X). For the latter organisms, improvements are most pronounced.

Different methods are suited for different data types. As you observed (Table 3), in the four real metagenomic sequencing datasets, the assembly results were best in most cases after F-HERO correction, whereas in these diploid or polyploid genomes (Table 6), R-HERO performed optimally in most cases.

### Polishing assembly results

It reflects common practice in TGS-based assembly to first assemble the raw reads, and to get rid of errors from the resulting contigs only thereafter, commonly referred to as "polishing assemblies." Here, we carried out experiments that allowed us to compare the quality of the resulting contigs with contigs that one obtains by applying error correction first (in particular with HERO) and assembling the corrected reads thereafter. When polishing assemblies, it is further common practice to make use of Racon [38] for polishing assemblies, because Racon explicitly suggests this practice. It is further current practice to use Canu for assembling the raw TGS reads. To obtain meaningful comparisons, we applied this practice to the three real metagenomic sequencing datasets and compared them with the original workflow of first correcting errors and then assembling corrected reads; see Table 3 (note

**Table 7** Assembling the corrected reads using HiFiasm-meta. Evaluating the effects of different correcting strategies on diploid/polyploid genome sequencing data. In the completeness (%) multicolumn, all is the k-mer completeness for both haplotypes combined, matis maternal hap-mer completeness and pat is paternal hap-mer completeness. Switch error rate = the ratio of the number of switches between true and alternative haplotypes to the total number of variant positions

| Assembler | Data | QV | Error rate (%) | Completeness (%) | | | Switch error rate (%) |
|---|---|---|---|---|---|---|---|
| | | | | **All** | **Mat** | **Pat** | |
| *Arabidopsis thaliana* | | | | | | | |
| Hifiasm-meta | Ratatosk | 23.55 | 0.44 | 35.31 | 16.77 | 16.64 | 10.49 |
| Hifiasm-meta | R-HERO | 22.71 | 0.54 | 36.77 | 18.06 | 18.45 | 10.58 |
| Hifiasm-meta | FMLRC | 33.93 | 0.04 | 8.36 | 3.32 | 4.55 | 7.58 |
| Hifiasm-meta | F-HERO | 24.49 | 0.36 | 19.23 | 8.58 | 9.44 | 8.80 |
| Hifiasm-meta | LoRDEC | 23.06 | 0.49 | 34.18 | 17.95 | 18.66 | 7.34 |
| Hifiasm-meta | L-HERO | 24.75 | 0.34 | 44.04 | 25.99 | 25.28 | 6.78 |
| *Arctia plantaginis* | | | | | | | |
| Hifiasm-meta | Ratatosk | 20.51 | 0.89 | 8.45 | 0.05 | 0.04 | 53.60 |
| Hifiasm-meta | R-HERO | 21.30 | 0.74 | 11.77 | 0.15 | 0.15 | 50.82 |
| Hifiasm-meta | FMLRC | 23.28 | 0.47 | 0.22 | 0.01 | 0.00 | 46.67 |
| Hifiasm-meta | F-HERO | 22.33 | 0.59 | 49.65 | 2.55 | 2.50 | 53.27 |
| Hifiasm-meta | LoRDEC | 16.40 | 2.29 | 4.04 | 0.07 | 0.07 | 49.91 |
| Hifiasm-meta | L-HERO | 21.70 | 0.68 | 48.35 | 2.63 | 2.57 | 52.33 |
| *Oryza sativa japonica* | | | | | | | |
| Hifiasm-meta | Ratatosk | 22.46 | 0.57 | 11.06 | — | — | — |
| Hifiasm-meta | R-HERO | 25.84 | 0.26 | 26.59 | — | — | — |
| Hifiasm-meta | FMLRC | 23.28 | 0.47 | 4.23 | — | — | — |
| Hifiasm-meta | F-HERO | 27.31 | 0.19 | 27.25 | — | — | — |
| Hifiasm-meta | LoRDEC | 21.42 | 0.72 | 19.91 | — | — | — |
| Hifiasm-meta | L-HERO | 24.23 | 0.38 | 39.96 | — | — | — |
| *Oryza sativa aromatic* | | | | | | | |
| Hifiasm-meta | Ratatosk | 21.88 | 0.65 | 2.26 | — | — | — |
| Hifiasm-meta | R-HERO | 23.93 | 0.41 | 22.95 | — | — | — |
| Hifiasm-meta | FMLRC | 19.75 | 1.06 | 3.10 | — | — | — |
| Hifiasm-meta | F-HERO | 25.30 | 0.30 | 31.03 | — | — | — |
| Hifiasm-meta | LoRDEC | 18.62 | 1.37 | 7.19 | — | — | — |
| Hifiasm-meta | L-HERO | 23.74 | 0.42 | 37.20 | — | — | — |
| *Fragaria x ananassa* | | | | | | | |
| Hifiasm-meta | Ratatosk | 21.35 | 0.73 | 75.29 | — | — | — |
| Hifiasm-meta | R-HERO | 23.14 | 0.49 | 81.44 | — | — | — |
| Hifiasm-meta | FMLRC | 19.60 | 1.10 | 21.53 | — | — | — |
| Hifiasm-meta | F-HERO | 23.14 | 0.49 | 42.45 | — | — | — |
| Hifiasm-meta | LoRDEC | 19.77 | 1.06 | 49.73 | — | — | — |
| Hifiasm-meta | L-HERO | 22.69 | 0.54 | 60.10 | — | — | — |

that first correcting errors enables one to use HiCanu instead of Canu, as a general advantage of this order). Because the principle of polishing contigs analogous to correcting errors in read, we also ran HERO on the contigs assembled by Canu from the raw reads; this way, we were in position to compare Racon (which was explicitly

designed for that) and HERO (not explicitly designed for polishing contigs) in terms of the qualities as tools by which to correct errors in contigs resulting from raw read-based assemblies.

See Additional file 1: Table S5 for results about polishing assemblies. Although Racon can reduce indel errors rates, it does not increase genome fraction and, still, exhibits greater indel error rates with other protocol, results of which are shown in the already above-mentioned Table 3. In addition to this disadvantage, Racon introduces new mismatch errors. In contrast, polishing with HERO achieves indel error rates that are on a par with those of Racon, without increasing the mismatch error rates, but rather decreasing them further.

To investigate why Racon increases the mismatch error rates, which supposedly reflects "overcorrection" (by mistankenly cleaning haplotype/strain-specific variation from certain haplotypes/strains), we visualized the differences between the results of either Racon or HERO polishing the Mock12 PacBio assemblies, using the IGV (Additional file 1: Fig. S4 and Table S5). While their workflows are similar overall, that is aligning short reads to contigs and then correcting the contigs (via establishment of a POA graph, so virtually an OG graph), HERO achieves favorable results. The obvious reason is the fact that HERO, unlike Racon, filters out reads from other strains based on SNP information (Step 2 in Fig. 1, see also Fig. 2), thus avoiding over-correction and introduction of new mismatches.

We also used Racon for correcting reads, so as to compare Racon's qualities of an error correction method for correcting errors in raw reads with the qualities of HERO. To not introduce any biases, we followed the exact same protocol for both Racon and HERO: we first corrected the TGS reads three times with Ratatosk, and then corrected one time with Racon and HERO, respectively (we did not apply Racon in an interative manner, as it has not been designed for such usage, but rather used HERO the way Racon was designed). Also in this protocol, as forecast, Racon introduces new mismatch errors into TGS reads, again an indication of overcorrection. Furthermore, also, the indel error rates exceeded those of HERO; see Additional file 1: Table S4 for these results. For the Bmock12 ONT and Bmock12 PacBio datasets in particular, which enjoy relatively low mismatch rates already before correction, Racon increased the mismatch error rates by one to two times.

### Runtime and memory usage evaluation

We evaluated the performance of runtime and peak memory of all methods on the dataset containing the 3 *Salmonella* strains, on a x86_64 GNU/Linux machine with 48 CPUs. The data volume of the NGS (Illumina) reads amounted to 296MB and the volume of the Pacbio CLR reads amounted to 281MB. Additional file 1: Table S2 reports CPU times and peak memory usages of the different HEC methods. Without any doubt, FMLRC is the fastest tool (likely thanks to the efficiency of the FM-index it makes use of): it only takes 0.29 h and 0.35 GB memory. Although the OG stage of HERO (1.54 h) is not as fast as FMLRC even though we utilize a divide and conquer strategy, it is still slightly faster than Ratatosk (1.79 h) and LoRDEC (2.40 h).

## Discussion

We have presented HERO, as an approach to correct errors in long, third-generation sequencing reads guided by additional usage of short next-generation sequencing reads. As such, HERO belongs to the class of hybrid error correction approaches for TGS reads. The major goal of HERO has been to correct errors in reads stemming from closely related genomes without getting confused by the similarity of the genomes during the correction process: the challenge is to not mistake phase-specific variants for errors and vice versa. This type of phase-induced confusion had been an issue that prior state-of-the-art approaches had not been able to overcome so far.

As we have demonstrated in experiments of great variability on datasets adhering to the currently highest standards, HERO has been able to successfully address this issue. The most prevalent, relevant such scenario are metagenomes, because they commonly contain various (most importantly bacterial) species, each of which exhibits multiple relevant strains. Because all such strains belong to the same species, their genomes (henceforth: haplotypes) tend to differ only by small amounts of variants. It is common that the frequency at which such haplotype-specific variants occur (1–2%, or even below) is below the sequencing error rate that commonly affects the classes of the longest TGS reads (5–15%). Relevant scenarios also arise in plant genomics, where polyploid genomes frequently occur. Unlike for metagenomes, one can usually assume that ploidy and coverage rates are known, which renders the task of error correction slightly less challenging, from a merely technical point of view. Our approach caters to correcting errors in reads derived from polyploid genomes just as well as from metagenomes.

Unlike the prior approaches, HERO is able to distinguish between such haplotype-specific variants on the one hand and errors on the other hand and to correct only the latter. As a consequence, HERO outperforms the state-of-the-art approaches by large margins, in particular with respect to the correction of artificial insertions and deletions (the most common type of error affecting TGS reads). Results further have shown that HERO's advantages increase on increasing similarity of the haplotypes and on decreasing sequencing coverage. Both of that is particularly favorable in mixed sample settings such as metagenomes, first and foremost. Of course, these properties are also important when dealing with polyploid genomes that are sequenced at relatively low coverage.

To the best of our understanding, we have therefore presented a clear novelty: haplotype-aware correction of errors in metagenomes, or other samples that exhibit a mix of closely related genomes/haplotypes, such as organisms of higher ploidy, had not been possible before. HERO is the first approach to address this successfully.

In this vein, it is important to realize that assembling the individual genomes making part of a metagenome (or also in certain polyploid genomes) in a haplotype-aware manner had been remaining a major challenge in the area of de novo genome assembly. Additional experiments of ours addressing this challenge provide clues to the striking benefits that HERO-corrected TGS reads offer for strain-aware metagenome assembly in particular.

Key to success for addressing these challenges successfully has been the insight that making use of two complementary classes of reads (TGS are long, but contain many errors, whereas NGS reads are short, but contain only little errors) also requires complementary read processing methodology. It is common knowledge that de Bruijn graphs

(DBGs) work particularly favorably when treating NGS reads. On the other hand, overlap-based data structures, such as multiple full-length alignments or overlap graphs (OGs), are the superior data structures to exploit the long-range dependencies that TGS reads capture. Remarkably, however, no earlier approach had been attempting to make use of both DBGs and multiple alignments/overlap-based structures. Note that, just as TGS and NGS reads are complementary by their properties, DBGs and overlap-based structures (OGs in particular) are complementary as per their characteristics as data structures. While DBGs break reads into $k$-mers (where $k$ is usually on the order of magnitude of the length of NGS reads), which successfully captures the redundancies of NGS datasets, overlap-based arrangements do not break reads into smaller pieces but preserve the information of the full length reads. The latter particularly caters to the fact that TGS reads are substantially longer than NGS reads.

The methological novelty of HERO has been to make use of both DBGs and overlaps/full-length alignments, just as well as of both TGS and NGS reads. Because NGS and TGS reads complement each other, and DBGs and full-length alignments/OGs complement each other, HERO is hybrid with respect to both usage of data and employment of data structures. In other words, HERO is double hybrid, which we have been referring to as "hybrid-hybrid."

Considering the variability of our experiments in a bit more detail, we have benchmarked HERO on both simulated and real data, reflecting scenarios referring to different bacteria, different numbers of strains, different relative frequencies of haplotypes (i.e., strains for example), and, last but not least, considering TGS reads from the two major sequencing platforms, PacBio and ONT. Moreover, we have considered various diploid and polyploid plant genomes. Results have demonstrated that HERO suppresses the indel and mismatch error rate substantially (by on average 60% and 20%, respectively, on the metagenome datasets, for example) in comparison with the state of the art. In other words, HERO more than halves the amount of indel errors in metagenomes that remain when using prior state-of-the-art approaches; it is important to know that indel errors are the predominant type of error that affects TGS reads. In summary, HERO has proven to be the obviously superior approach. As a possible suggestion for a practitioner who is interested in the application of our protocols, we recommend to use the "R-HERO" protocol for scenarios of known ploidy (based on the fact that this protocol appears to optimally preserve the haplotype-specific variation) and the "F-HERO' protocol for metagenomes, in line with the original intentions of the DBG-based methods (Ratatosk and FMLRC) involved in them.

An immediate interpretation of these (in several benchmark scenarios arguably even drastic) advantages is the consideration that capturing haplotype specific mutation already during error correction is not just beneficial but perhaps even imperative when seeking to remove errors from TGS reads for various purposes. A hint that underscores this thought is the fact that strain-aware assemblies of metagenomes improved significantly when using TGS reads that were corrected in a haplotype-aware manner. Evidently, trying to capture the differences between haplotypes only during assembly, that is, only after the correction stage comes too late. HERO therefore corroborates the rather recent idea that distinguishing between haplotypes already during the very first stage of read processing, which usually reflects the correction of the sequencing errors

they harbor, can be a must. The insight is complemented by experiments that compare the practice of first assembling raw reads and only then correcting errors in the resulting assembled contigs, with the most natural workflow, namely correcting errors first, and then assembling corrected reads. The results support one's feeling that correcting reads before assembly is the more natural, so generally preferable workflow.

A further potentially remarkable observation has been the fact that HERO corrected TGS reads enable us to make use of assembly tools developed for PacBio HiFi reads (here: HiCanu); in fact, the error rates affecting HERO corrected reads are substantially below the ones affecting PacBio HiFi reads. Combining this with the fact that PacBio CLR and ONT reads are on average three to five times longer than PacBio HiFi reads provides one with a protocol that is not only substantially cheaper but also has substantial advantages because of the significantly longer TGS reads. From that point of view, HERO has presented itself as an inexpensive and, from relevant points of view, also superior alternative for laboratories that have been equipped with both NGS and TGS platforms already. The advantages in terms of costs and read length establish reasonable compensation for the lack of convenience one has to bring up when designing experiments involving both TGS and NGS reads.

Of course, future improvements are conceivable: in particular, while not subject to excessive requirements, HERO does not outperform the state of the art with respect to usage of computational resources. For larger datasets in particular, although still being perfectly feasible in practice, HERO experiences greater peak memory and runtime requirements. We recall that HERO makes use of DBGs and MAs/OGs as data structures, where OGs in particular have proven to be prone to larger computational demands if not implemented carefully. HERO already addresses this issue, which leads to the already fairly well-behaved runtimes and memory requirements. However, there is hope for further improvements in this regard. Note that, at one point, HERO aligns NGS reads with (DBG-based pre-corrected) TGS reads and then uses the TGS read as a template for phasing the NGS reads that align with them and subsequently discarding the NGS reads that do not agree with the TGS template read in terms of phase. This TGS template-based phasing procedure consumes the largest amount of time. A conceivable solution is to pre-phase the long TGS reads prior to aligning them with the NGS reads. If pre-phased sufficiently well, TGS reads get aligned only with NGS reads that stem from the same phase, which avoids the time consuming filtering out of spurious NGS-TGS alignments. Complementing the pre-phasing with enhanced sequence-to-sequence alignment techniques has the potential to yield a hybrid sequencing error correction method that is superior in terms of runtime, in addition to being superior in terms of removal of errors (which HERO already is).

## Conclusions

We have presented HERO for correcting errors in long, TGS reads. HERO particularly addresses scenarios characterized by multiple haplotypes, such as di-/polyploid and metagenomes. Unlike prior approaches, HERO does not mistake low-frequency, haplotype/strain-specific variations for errors, so it preserves the haplotype identity of the reads during correction. In such scenarios, HERO outperforms extant approaches by large margins, suppressing error rates by a factor of up to 10 in the most drastic cases.

Therefore, HERO particularly caters to genome assembly protocols where strain or haplotype awareness is key.

HERO makes use of short, NGS read data to aid in the correction procedure. For optimal exploitation of the characteristic properties of TGS reads on the one hand, and NGS reads on the other hand, HERO employs both de Bruijn and overlap graph-based strategies. HERO is "hybrid-hybrid" insofar as it is hybrid in terms of both sequencing protocol and data structure usage.

## Methods

### Simulated data

To compare the performance of different approaches, we used CAMISIM [42] (version 0.0.6) to generate three simulated hybrid sequencing datasets (Illumina MiSeq and PacBio CLR) containing 3 *Salmonella* strains, 20 bacterial strains (10 species), and 100 bacterial strains (30 species) respectively. CAMISIM is a popular metagenome simulator which can model the second- and third-generation sequencing data at different abundances and multi-sample time series based on real strain level diversity.

The length of the simulated Illumina MiSeq reads is 2X250 bp, at an insert size of 450 bp. The N50 of PacBio CLR reads is 10 kbp at an average sequencing error rate of 10%. As per the principles of CAMISIM, the abundance of different strains is uneven, as sampled from a log-normal distribution. The average coverage of both Illumina MiSeq and PacBio CLR data for the three simulated communities is 10X. The genomes of the 3 *Salmonella* strains were obtained from earlier work [43]. The genomes for the 20 bacterial strains and the 100 bacterial strains communities were downloaded from an earlier study [44]. For details with respect to Genome IDs and their average nucleotide identity (ANI), please see Additional file 1: Table S1.

Furthermore, in order to evaluate the influence of the coverage of short reads, we produced 6 sequencing datasets that mix simulated with real data; this reflects a common simulation scenario and is referred to as "spike-in" data. The idea is to evaluate how methods correct the simulated ("spiked-in") data—for which one knows the ground truth—as part of a scenario that is as real as possible otherwise (but of course lacks ground truth for the real data, which explains the need to spike in simulated reads). In more detail, we spiked 6 different, real gut metagenome sequencing datasets, resulting from experiments referring to identify functional characterization of low-abundance and uncultured species in the human gut [45] (project number: PRJNA602101) with simulated reads from 10 well-known *Salmonella* strains, as downloaded from [43]. For simulating reads from the *Salmonella* strains, we again made use of the CAMISIM simulator. By following the properties of the real sequencing data (so as to have optimally comparable results), the length of the synthesized short reads is 2X150 bp. To account for the influence of read coverage, the coverage of synthesized short reads for the spiked-in strains ranged from 5X to 30X, at steps of 5X, across the 6 real datasets, that is, each of the 6 spiked-in real datasets refers to one particular level of coverage for short reads. Instead, the coverage of the simulated TGS reads remains fixed at 10X across the 6 real datasets. For the sake of a less runtime-intense evaluation in the light of the large amount of reads, we randomly extracted 16,333,444 NGS reads and 181,092 TGS reads from per real hybrid sequencing data, and further processed only these. For details in

terms of 10 *Salmonella* Genome IDs and SRA identifiers, please see Additional file 1: Table S1, "spike-in *Salmonella*."

### Real data

We considered two microbial communities for which both TGS and NGS data were available in our experiments:

*Bmock12* is a mock community that contains 12 bacterial strains from 10 species [46]. The mock community was sequenced using all ONT MinION, PacBio, and Illumina sequencing platforms. We downloaded the corresponding datasets from SRA (illumina SRR8073716, ONT SRR8351023, PacBio SRR8073714). The N50 of the length of the ONT and PacBio reads is 22772 and 8701, respectively. The read length of the Illumina reads is $2 \times 150$ bp (average insert size 302.7 bp). Note that the number of reads that mapped to *Micromonospora coxensis*, as one of the 12 strains, was negligible [46]. So, one virtually deals with only 11 bacterial strains. The average coverage of the corresponding 11 strains ranges from 74.56X to 3093.79X (median: 1376.35X). For the sake of a less runtime-intense evaluation in the light of the large amount of duplicates among the reads, we randomly extracted 20% of the reads and further processed only these. Finally, note that challenges of this dataset are to correct the long reads of the two species whose strains exhibit the greates average nucleotide identities (ANIs). This concerns *Marinobacter* species and the *Halomonas* species in particular, because these species contain pairs of strains characterized by 85% and 99% ANI, respectively.

The second microbial community drawn from natural whey starter cultures (*NWCs*) [47]. The NWC metagenome samples were sequenced using Illumina MiSeq at a read length of $2 \times 300$ bp and, in addition, using PacBio and ONT. We downloaded sequence datasets from SRA (Illumina SRR7585899 and SRR7589561, ONT SRR7585900, PacBio SRR7589560). The N50 of the TGS PacBio and ONT read length is similar, being 11895 and 9562 respectively. In an earlier study, complete genomes for 6 bacterial strains from 3 species were obtained, by means of running a hybrid assembly method [47]. Genbank numbers of the corresponding 6 genomes are CP029252.1, CP031021.1, CP031024.1, CP031025.1, CP029252.1, and CP031021.1; we used these assembled genomes as ground truth when evaluating corrected long reads.

*Arabidopsis thaliana* is a classic plant model organism [48]. We downloaded its sequencing data from SRA (Illumina: SRR3703082; PacBio: SRR3405292, SRR3405294, SRR3405296, SRR3405305, SRR3405307). The length of the Illumina reads is $2 \times 250$ bp. For PacBio, we filtered out reads of length less than 5000 bp, which resulted in 327,180 reads overall, with the size of the final fasta file amounting to 5.2G, which is equivalent to 39X coverage (the Arabidopsis thaliana reference genome is 136 Mbp long). Additionally, we directly downloaded the hapmer data for its two parents (col0 and cvi0) from the Merqury github [39] to evaluate the switch error rate.

*Arctia plantaginis*, the wood tiger moth, is a diploid insect. We obtained the corresponding trio sequencing data from a previous study [49]. The corresponding sequencing data was downloaded from SRA (Paternal Illumina: ERR3890308; Maternal Illumina: ERR3890309; F1 Illumina: ERR3909542; F1 PacBio: ERR3890511 and ERR3890512). The length of the Illumina reads is $2 \times 150$ bp. For PacBio, again we discarded reads of length shorter than 5000 bp, resulting in 658,434 reads, yielding a fasta file of 10G, equivalent

to 18X coverage (the reference genome is 559Mbp). We built a database with Merqury using the Illumina data from the two parents to prepare for evaluating switch errors in the following.

*Oryza sativa japonica* is a type of rice, representing one of the most important staple foods worldwide. The sequencing data was obtained from a previous study [50] (Illumina: SRR20046022; ONT: SRR20046021). The reads length is $2 \times 150$ bp. From the ONT reads, we again only kept reads of length at least 5000 bp, resulting in 146,150 reads, yielding a fasta file of 4.1G, equivalent to 11X coverage (the reference genome is 366 Mbp).

*Oryza sativa aromatic* is also a rice cultivar. The sequencing data was obtained from another earlier study [51] (Illumina: SRR10302329; PacBio: SRR10302165 and SRR10302169). The Illumina reads are $2 \times 150$ bp; PacBio reads of length at most 5000 bp were discarded, resulting in 477,456 reads, amounting to a fasta file of 7.6G, corresponding to 21X coverage (reference genome is 368 Mbp).

*Mangifera indica* is a tetraploid mango. The sequencing data is from [52] (Illumina: SRR8281984; PacBio: SRR11050202). The Illumina reads are of length $2 \times 300$ bp, and (as usual) only PacBio reads longer than 5000 bp were retained, totaling 724,948 reads. The corresponding fasta file was of size 15G, corresponding to 40X coverage (the reference genome is 379 Mbp).

### Workflow: detailed description of steps
#### DBG stage: removal of artificial indels
We first applied the DBG-based HEC methods Ratatosk [15], FMLRC [14], and LoR-DEC [13] for reducing the artificial indel content in the TGS reads, so as to improve the quality of (overlap based) TGS-NGS alignments to be computed subsequently. Note that these methods have proven to establish the current state of the art. We recall further that attempts of ours to reach improvements for our particular purposes (improving TGS-NGS alignments in the first place) had no effects. This explains why we used these methods without further modifications.

However, we found that repeated application of the DBG-based methods lead to further improvements of the TGS-NGS alignments. Moreover, repeated application of these methods lead to small, but still noticeable improvements in terms of solely DBG-based HEC (which may have passed unnoticed by the authors of the respective methods). For optimal usage of the DBG methods in our framework, we found that results kept improving on increasing numbers of iterations; however, after three iterations, improvements became negligibly small. Therefore, for optimized results, we suggest using these methods three times prior to the MA-based stage of HEC. If not further specified, our results refer to threefold application of the DBG-based pre-correction.

#### OG stage: distinguishing errors from haplotype specific variation
Step 1: Calculating the overlap between NGS and TGS reads

See step 1 in Fig. 1. Relying on the improved quality of alignments of the TGS reads with other reads thanks to the removal of the majority of disturbing indel artifacts, we aligned each of the NGS reads on the one hand with each of the TGS reads on

the other hand using Minimap2 [19]. The result of these TGS-NGS read alignments is a construction that aligns a TGS read with all NGS reads that overlap it on the one hand.

Based on this overlap-based pile of NGS reads, for which the target TGS reads provides coordinates, one now can aim to identify NGS reads that are not in phase with the TGS target read, which need to be removed from further consideration.

Step 2: Phasing and filtering out NGS reads

See step 2 in Figs. 1 and 2 for illustrations in the following. The goal is to keep only NGS reads that stem from the haplotype of the TGS target read or, in other words, that are "in phase with the TGS read," while discarding (filtering out) all others. Since the overlap-based pile of NGS reads spans information across the entire length of the TGS read, one can make this distinction in a decisively enhanced way. Upon having discarded NGS reads not in phase with the TGS read, errors can be corrected in a superior way in subsequent steps.

To identify SNPs that are characteristic of haplotypes the TGS read does not stem from, we inspect the mismatches contained in the NGS-TGS alignments one by one. In practical terms, we screen the information provided through the alignments computed by Minimap2. Note that the output of Minimap2 contains exact information about mismatches and indels in the corresponding alignments. Three different, relevant scenarios can arise; see Fig. 2 for an illustration of the following arguments.

*Scenario 1*: "All" NGS reads disagree with the TGS read on a particular position; see "P1:X" in Fig. 2. The only conclusive explanation for this scenario to arise is the occurrence of an error in the TGS read. To account for possible sequencing errors in the NGS reads that coincidentally match the error in the TGS read, "all" is defined to be at least five NGS reads that agree with the nucleotide in the TGS read.

*Scenario 2*: At least five NGS reads agree with the nucleotide in the TGS read, whereas at least five NGS reads disagree with it; see "P2:X" in Fig. 2. In that case, we conclude that the nucleotide in the TGS read is correct based on the sufficient support provided by NGS reads. On the other hand, we conclude that the sufficiently large amount of NGS reads that support the existence of a nucleotide that differs from the one showing in the TGS read gives rise to a SNP that characterizes a phase that differs from that of the TGS read. Subsequently, we discard (filter out) all NGS reads that exhibit such a SNP not in phase with the TGS read from the alignment pile.

*Scenario 3*: Nucleotides disagreeing with the one of the TGS read that are supported by at most four NGS reads; see "P3:X" in Fig. 2. We conclude that the nucleotides in the corresponding NGS reads are due to sequencing errors. Note that we do not discard the corresponding NGS reads but continue to use them in subsequent steps, because we decided that they stem from the phase of the TGS read, which apart from that particular position contain valuable information on sequencing errors in other positions of the TGS read.

Note that we refrain from correcting errors in the TGS read at this point, although Scenario 1 already reflects that we are rather certain to have spotted errors in the TGS read. Instead, we now re-examine the alignment pile upon having discarded all NGS reads that stem from a haplotype that does not match the haplotype of the TGS read. Rephrasing this, we can now work with an OG all nodes of which stem from the same

phase—relying on this now well justified assumption enhances the identification of errors in the TGS reads decisively.

Step 3: Window segmentation

See step 3 in Fig. 1 for the following. After removal of non-matching NGS reads, we segment the alignment pile into smaller windows. We do this for two reasons.

First, NGS coverage can be subject to substantial fluctuations along the genomic region that the (much longer) TGS read spans. The volatility in terms of coverage can prevent accurate calculations that are necessary for the identification of errors in the TGS reads relative to the coverage with NGS reads. Because coverage varies subtantially less in smaller regions, segmentation of the alignment pile is helpful.

Second, calculating coverages and SNP content for each position in a TGS read is a computationally expensive operation. In that respect, segmentation supports the parallelization of this operation. This speeds up runtimes considerably, as one can run the necessary computations for each segment window in a separate thread.

To support fast execution of computations further, we only carried out a more detailed analysis of a locus if there were both NGS reads supporting the nucleotides of the TGS reads and NGS reads not supporting it. If all NGS reads disagreed with the TGS reads, we concluded that the TGS read carries an error and correct it to the nucleotide shown by the NGS reads; in turn, we omitted any kind of action on a position where all NGS reads agreed with the TGS read.

In the following, we will refer to the part of the target TGS read that corresponds with one of the segments as *target subread*.

Step 4: Correcting errors in target subreads

See step 4 in Fig. 1 for an illustration. For each segment of the alignment pile, we consider the TGS target subread on the one hand, and all parts of the NGS reads that align with it. That is, for a particular window, we do neither consider the parts of NGS reads that do not make part of the window, nor do we consider NGS reads at their full length if none of their parts makes part of the window.

For each window, considering the corresponding TGS target subread and the (parts of the) NGS reads that align with it, we compute a partial order alignment (POA) as per the algorithm provided by [53]—here we run Racon [38], which has implemented exactly that algorithm. Note that this POA does no longer consider the target TGS subread. Rather, it computes a particular, graph-based multiple alignment of the NGS read segments in that window. Based on the resulting POA, we generate a consensus sequence for the window in question. The operation of generating consensus sequence corresponds to turning the POA graph into an OG, which is an immediate operation because the POA graph spells out overlaps of NGS read segments (the segments correspond to subpaths in the POA graph). This virtual OG is then traversed to assemble the NGS read segments into a consensus sequence segment. By the consideration that the consensus sequence is the correct sequence of the haplotype that one is dealing with, one can now correct errors in the target subread, namely by adapting nucleotides in the target subread to the ones of the consensus sequence, if not agreeing with it, or, in other words, by replacing the target subread by the consensus as the corrected version of it.

As a technical detail, POAs are computed by making use of a single instruction/multiple data (SIMD) type of implementation, as implemented in Racon [38], which we use.

This optimally supports parallel processing. In addition, this also warrants high accuracy of the resulting consensus further.

Step 5: Concatenating corrected target subreads

See step 5 in Fig. 1: corrected target subreads are concatenated to the original full-length, but now corrected target TGS read. Here, concatenation just corresponds to the straightforward idea of "patching together" target subreads in the order corresponding to their original order before correction.

Step 6: Merging corrected target reads

See step 6 in Fig. 1. Step 6 just corresponds to outputting the now corrected full-length reads for further usage. Note that further usage does not necessarily refer to usage in downstream applications but may mean that corrected TGS read undergo another cycle of correction, that is, serve as input TGS reads for repeated application of steps 1–6. As supported by respective experiments, the default is to run through steps 1–6 three times, before outputting corrected full-length reads for their application in experiments/computations that depend on clean, error-free TGS reads. We recall that haplotype-aware de novo assembly is likely the most fundamental such application.

### Evaluation criteria

#### *Metagenomes*

In the evaluation, we considered all relevant categories, as output by MetaQUAST V5.1.0 [54], as a prominent assembly evaluation tool, for the metagenome datasets, for all of which applicable haplotype/strain-resolved reference genomes were available. Based on general recommendations, we added the flags –ambiguity-usage all and –ambiguity-score 0.9999 when evaluating metagenomic data. All other parameters reflect default values. In the following, we briefly define the metrics we are considering when using QUAST. For more detailed explanations, see http://quast.sourceforge.net/docs/manual.html.

`Indels/100 kbp`

The sequence generated by raw Pacbio CLR and ONT reads is prone to contain a large amount of indel errors. Here, indels per 100 kbp refers to the average number of insertion or deletion errors per 100,000 aligned bases in the reads.

`Mismatches/100 kbp`

This reflects the average number of mismatch errors per 100,000 aligned bases in the evaluated reads.

`N/100 kbp`

This denotes the average number of uncalled bases (Ns) per 100,000 bases in the reads subject to evaluation.

`Genome fraction (GF)`

GF is the percentage of aligned bases in the reference genome against which reads become aligned. In other words, GF reflects how much of each of the strain-specific genomes can be covered by the evaluated reads.

`Misassembled contigs (MC)`

If a read involves at least one misassembly event, it is counted as *misassembled read.* A misassembly event is defined to reflect that reads align against the true sequence while having a gap of more than 1 kpb, or showing an overlap of more than 1 kbp with

a different strand, or even a different strain. We report the percentage of misassembled reads relative to the overall number of evaluated reads as "misassembled reads."

`NGA50`

NGA50 is defined to be maximum length of a read such that all reads of that length or longer cover at least 50% of the true sequence by their alignments with it.

### *Plant genomes*

When evaluating the plant genome datasets, we considered Merqury [39] for evaluating results, which is an approved tool for scenarios where applicable (in particular haplotype-resolved) reference genomes are missing. We briefly define the metrics when using Merqury. For more detailed explanations, see https://github.com/marbl/merqury.

`Completeness`

The k-mer completeness measures the proportion of reliable k-mers in the read set that are also present in the assembly. It indicates to what degree the assembled genome contains all the genes and sequences from the original DNA sample. From another point of view, completeness accounts for the haplotype awareness of the assembly.

`Error rate`

In Merqury, the error rate refers to the combined sum of indel and mismatch error rates.

`QV`

Merqury also utilizes k-mers to estimate the frequency of consensus error in the assembly, thereby evaluating the quality of the assembled contigs.

`Switch error rate`

The rate of the number of switches between true and alternative haplotypes to the total number of variant positions.

## Supplementary Information

---

**Additional file 1.** Supplement. This contains all supplementary materials referenced in the main text.

**Additional file 2.** Supplementary Tables.

**Additional file 3.** Review history.

---

**Acknowledgements**
Not applicable.

**Review history**
The review history is available as Additional file 3.

**Peer review information**

**Authors' contributions**
XK and AS developed the method. XL, XK, and AS wrote the manuscript. XK, XL, and JX conducted the data analysis. XK implemented the software. All authors read and approved the final version of the manuscript.

**Authors' Twitter handles**
@XiongbinK (Xiongbin Kang); @ASchonhuth (Alexander Schönhuth).

**Funding**
 XL and JX are supported by the Fundamental Research Funds for the Central Universities (541109030062). AS received funding from the European Union's Horizon 2020 research and innovation programme under Marie Skłodowska-Curie grant agreements No 956229 (ALPACA) and No 872539 (PANGAIA).

### Availability of data and materials

The source code of HERO is GPL-3.0 licensed and publicly available at [55]. The code for reproducing the results in the paper is deposited in Code Ocean with the capsule [56].

The corresponding datasets were obtained from the Sequence Read Archive (SRA) database (Bmock12: Illumina SRR8073716, ONT SRR8351023, PacBio SRR8073714; NWCs: illumina SRR7585899 and SRR7589561, ONT SRR7585900, PacBio SRR7589560. Arabidopsis thaliana: Illumina SRR3703082, PacBio SRR3405292, SRR3405294, SRR3405296, SRR3405305 and SRR3405307; Arctia plantaginis: Paternal Illumina ERR3890308, Maternal Illumina ERR3890309, F1 Illumina ERR3909542, F1 PacBio ERR3890511 and ERR3890512; Oryza sativa japonica: Illumina SRR20046022, ONT SRR20046021; Oryza sativa aromatic: Illumina SRR10302329, PacBio SRR10302165 and SRR10302169; Mangifera indica: Illumina SRR8281984, PacBio SRR11050202).

## Declarations

### Ethics approval and consent to participate
Not applicable.

### Consent for publication
Not applicable.

### Competing interests
The authors declare that they have no competing interests.

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

## 