## [**Additional file 3.** Review history. · Genome Biology]

Review History

First round of review

Reviewer 1

Were you able to assess all statistics in the manuscript, including the appropriateness of statistical tests used? Yes. No statistical tests are performed in this work.

Were you able to directly test the methods? No.

Comments to author:

Kang et al. "Hybrid-hybrid correction of errors in long reads". This work extends prior methods on read-based correction. One frequent concern is the "overcorrection" that tends to happen with mixed haplotype genomes - such as metagenomic or polyploid genomes. The proposed pipeline stitches together several off-the-self correction/consensus programs, as well as implements a novel method to correct long reads in a strain-aware or haplotype-aware manner. While the authors clearly had the metagenomics application in mind and demonstrated so in the paper, I would very much like to see a similar application in a diploid/polyploid genome. Oh well ...

Front and foremost, is the read-based correction still relevant in 2023? The error rate is already quite low for Revio HiFi (Q30) and ONT10.4.1 (Q24 to Q31). This level of error is on par with Illumina reads on mismatches but perhaps still lack a bit in indels. But the accuracy gap is closing rapidly. This brings into the question whether the DBG-based pre-correction is still necessary with Q30 TGS? The authors should at least discuss if the latest long read technologies might reduce or even deprecate the need for correction. In my area (large plant genome assemblies), the current SOTA is to NOT correct in anyway and in many cases HiFi reads are sufficient to put together a highly contiguous, in some cases, gap-free genomes nowadays. Additionally, how does this compare to the approach of polishing the draft assemblies with NGS reads, i.e. running Racon directly on draft contigs to fix indel errors, which is also a common practice? Would it make sense to do a comparison here in this paper?

Large complex polyploid genomes may contain transposons which makes over-correction much more likely in the DBG step. Large metagenomics samples may be too difficult as shown in very high mis-assembled contigs, as measured in MC% in Table 3. The easier cases, on the other hand, may be straightforward whether you correct or not - so I really wonder if the method might just have a narrow window of applicability, or is it expected to work generally well outside the datasets tested in this work? To be honest, I was a bit concerned by the performance listed in Table 3. Specifically, the MC% (fraction of misassembled contigs) seem quite high for many of the HERO versions. Indeed, the HERO versions have significantly higher MC% compared to non-HERO versions in all cases presented. Given these misassembled contigs, I felt that the improvement on error rates and NGA50 isn't nearly as impressive as the authors claimed.

The component that is the most novel and deserves the most in-depth analysis is the step where the non-specific alignments are recognized and removed. This step is expected to reduce "over-corrections", which the authors described well in the Background, which is indeed a pain point for many prior methods. The important question here - are overcorrections fully addressed by HERO? It is not clear from the paper, and it would be helpful to use a few real examples to build

up such an intuition and provide an estimated proportions of the overcorrections reduced due to this step.

Across the entire paper, it isn't clear which DBG implementation to use under which circumstances? The authors did not attempt to make recommendations among these choices. Are the users expected to try all three of FMLRC, LorDec, and Ratatosk? This is unclear. It might give the false impression that that there is always some improvement in the HERO-version using any of the three methods. In real projects, the users unfortunately won't have a ground truth to compare to, so it often is quite difficult to know which DBG method to pick. As seen in Table 3, the winner can be any of F-, L-, or R-HERO and the authors should provide a rough guide.

Another concern of mine is the time cost for HERO, which the authors acknowledged in Discussion - since there can be a few iterations and repetitions within each iteration- not to mention potentially run through all three DBG methods, F, L and R. To run the corrections for these many iterations/repetitions/methods may be prohibitive for larger metagenomics samples, or complex polyploid genomes. The authors did suggest that "TGS template based phasing procedure consumes the largest amount of time. A conceivable solution is to pre-phase the long TGS reads prior to aligning them with the NGS reads..." If the authors already have an idea for performance improvement, why not just implement it to address this bottleneck?

"The POA can accumulate artifacts if not confined to small genomic regions." Please show evidence of this and how the segmentation bin size or boundary is determined.

It may be my pet peeve, but the authors often use language in the paper that often sound more sophisticated than it really is. As an example, "As a technical detail, we computed POA's by making use of a single instruction / multiple data (SIMD) type of implementation ..." - it really is just invoking Racon - 'POA' and 'SIMD' is the implementation details of Racon - the authors should just spell out Racon in the methods. Similarly, the use of overlap graph (OG) in HERO is quite trivial and a bit misleading since it doesn't traverse the graph structure as in a typical genome assembly algorithm. In the current context, it is just referring to sorting the overlaps suggested by MINIMAP2 into short read piles. There is no need for a graph data structure and nothing more than a sort or hash table with long read as the key.

Lastly, the code quality on GitHub is poorer than I'd expect and does not seem to fully match some of the methods described in the text. Please also consider including more examples and documentations.

Minor:

- Background appears too verbose and seems to contain substantial overlap between the initial section and Related Work section.
- Explain R-Hero on its first occurrence (e.g. HERO corrected Ratatosk)
- Typo: "data structruess"

Reviewer 2

Were you able to assess all statistics in the manuscript, including the appropriateness of statistical tests used? Yes: No statistics review needed.

Were you able to directly test the methods? No.

Comments to author:

Overview:

The authors consider the hybrid correction of long TGS reads using short and accurate short reads. This problem has previously been approached by e.g. aligning short reads to the long reads or by aligning long reads to the de Bruijn graph (DBG) of the short reads. Here the authors propose to combine these two approaches into a pipeline where first the DBG based approach is used and then the short read alignment based approach (the author use the term overlap graph (OG) approach) is used to further correct the reads. The authors implement this pipeline in a tool called HERO and show improvements over the previous methods when reads are corrected by HERO. The improvements are most pronounced in metagenomic settings where the different strains closely resemble each other.

Originally DBG based approaches replaced the alignment based approaches because they were superior when it came to space and time efficiency. I find it interesting that the authors show here that when leveraging the reads already corrected by DBG approaches and the highly efficient read mapper developed after the first correction attempts, the alignment based approaches become competitive also in terms of efficiency.

The paper is well written and easy to understand.

Suggestions for revision:

1) The term alignment based approach has been used in previous literature to describe hybrid correction approaches relying on aligning short reads to the long reads. I do not see that it would be justified here to introduce a new term, OG based correction, for this approach.

2) Page 7: "L-HERO ... its indel error rate amounted to only 37.9% of that of LoRDEC only:" I believe this should be 37.9% lower than LoRDEC, not 37.9% of the error rate of LoRDEC.

3) Page 10: "HERO suppresses the indel and mismatch error rate by 60% and 20%, respectively, in comparison with the state of the art.": These results vary from one data set to another so this sentence should be reformulated to be more precise.

4) Page 12: "we aligned all NGS reads on the one hand with all TGS reads on the other hand using Minimap2.": Unclear sentence, please reformulate.

5) Some typos:

- Page 9: "...ONT data set, The performance"

- Page 10: "state-pf-the-art" (also just before state-of-the-art has been written without hyphens)

- Page 11: "data structruess"

We would like to thank the Reviewers for their helpful and fruitful input. We are convinced that addressing their points has helped considerably to improve the manuscript. In the following, please find a point-by-point rebuttal of comments, questions and suggestions.

Point-by-point rebuttal

Reviewer #1:

Comment: Kang et al. "Hybrid-hybrid correction of errors in long reads". This work extends prior methods on read-based correction. One frequent concern is the "overcorrection" that tends to happen with mixed haplotype genomes - such as metagenomic or polyploid genomes. The proposed pipeline stitches together several off-the-self correction/consensus programs, as well as implements a novel method to correct long reads in a strain-aware or haplotype-aware manner. While the authors clearly had the metagenomics application in mind and demonstrated so in the paper, I would very much like to see a similar application in a diploid/polyploid genome. Oh well ...

Response: We thank you for pointing out the application scenario of polyploid organisms, which is of great importance in plant genomics in particular. We fully agree that adding experiments on polyploid organisms widens the scope of our method in a highly relevant manner. Please note first that we had been generally aware of polyploid genomes as a relevant area of application, see earlier publications from our lab, e.g. [Baaijens et al., 2019, Bioinformatics].

The reason why we had been focusing on metagenomics here is the fact that scenarios characterized by unknown ploidy and uneven coverage across haplotypes / strains are the statistically most challenging ones. The uncertainty with respect to ploidy and coverage induces particular obstacles, such as coverage breaks and the necessary disentangling of reads into portions that agree with the participating species gives rise to enhanced challenges in metagenomics.

This paper has been a methodical attempt to push the limits in even the statistically/methodically most challenging scenarios when aiming to obtain the longest and cleanest reads possible. This explains why we put particular emphasis on metagenomics. Note that our particular expertise has been referring to related endeavors in the recent past already: see for example StrainXpress [NAR, 2022], Strainline [Genome Biology, 2022], VeChat [Nat. Comms., 2022]). In all these publications, we aimed to demonstrate how to push particular limits in metagenomics.

Nevertheless, again, we would like to thank you for pointing out that our analysis lacked diploid / polyploid plant genomes as another important, most relevant area of application. We certainly did not mean to neglect plant genomics as an important area of application—we were just a bit less challenged by the somewhat reduced level of difficulty induced by the enhanced level of certainty with respect to ploidy and coverage. Still we are aware that the length and the repeat content of plant genomes induces difficulties, where the repeat content even induces uncertainties that are related to the issues arising from the unknown ploidy in metagenomics.

In the revised version, we have now added data sets referring to the diploids *Arabidopsis thaliana*, *Arctia plantaginis*, *Oryza sativa aromatic* and *Oryza sativa japonica*. Further, we extended our analyses to the tetraploid mango (*Mangifera indica*), for all of which we were able to retrieve applicable data sets (trio data in particular, important for just sound evaluation of results). Results are collected into Table 5. On these five real data sets, we demonstrate that in direct comparison with FMLRC, Ratatosk and Lordec (as the only competitive approaches), we achieve reductions in sequencing error rates by on average 48.58% (23.4% -

71.6%). This is evidence that HERO also brings up substantial improvements also here, despite the fact that plant genome scenarios are presumably easier for the prior approaches, due to the increased degree of certainty inherent to statistics such as ploidy and coverage.

Comment: Front and foremost, is the read-based correction still relevant in 2023? The error rate is already quite low for Revio HiFi (Q30) and ONT10.4.1 (Q24 to Q31). This level of error is on par with Illumina reads on mismatches but perhaps still lack a bit in indels. But the accuracy gap is closing rapidly. This brings into the question whether the DBG-based pre-correction is still necessary with Q30 TGS? The authors should at least discuss if the latest long read technologies might reduce or even deprecate the need for correction. In my area (large plant genome assemblies), the current SOTA is to NOT correct in anyway and in many cases HiFi reads are sufficient to put together a highly contiguous, in some cases, gap-free genomes nowadays.

Response: Undeniably, the latest sequencing technologies / protocols such as PacBio HiFi (Q30) and ONT (Q20+/Q30+) bring great base accuracy to the table already in their raw form. Just as it was standard to not correct errors in Illumina reads, it is justified to process these new classes of third-generation sequencing reads without error correction prior to analysis, at least for various relevant scenarios of interest.

However, if one would like to “go for the stars”, and compute assemblies of optimal quality, in particular when it is about operating at strain/haplotype resolution, there are still excellent reasons to make use of hybrid approaches that integrate the earlier TGS reads with NGS reads.

Why is this?

Earlier classes of TGS reads (or also the latest Ultra ONT reads, which also harbor errors at rates >5%) are considerably longer than HiFi (also Q30) and ONT Q20+/Q30+, while NGS (Illumina) reads enjoy error rates even somewhat below of those of HiFi Q30 and ONT Q20+/Q30+, in particular when one deals with indels, as NGS reads harbor near-to-none indel errors (NGS is Q50+ wrt indels and approaches Q30 wrt mismatches, see e.g. doi.org/10.1186/s12859-016-0976-y).

This means that (earlier, very long, or also Ultra ONT type) TGS reads and NGS reads complement each other just perfectly. As we demonstrate here:

- 1) Error rates of TGS reads after hybrid error correction drop below the error rates of HiFi Q30 / ONT Q20+/Q30+ reads, even quite substantially
- 2) Since earlier generations of TGS reads are longer than Q20+/Q30+ reads (for example, PacBio CLR and PromethION reads are 3-5 times longer on average than HiFi reads [1,2]), hybrid correction of CLR/PromethION reads (for example) leaves one with reads that are (still by far) the longest and the cleanest.

The bottom line is that hybrid correction of the longest TGS reads may be just what is still necessary in various scenarios of application (in particular in metagenomics) while not strictly necessary in various other scenarios (where, for example, haplotype resolution is not the primary purpose).

The second reason why hybrid error correction could be of substantial interest are the costs of HiFi / ONT Q20+/Q30+ read sequencing, because these costs are prohibitive for the majority of sequencing laboratories worldwide. For example, regular (non-Revio) HiFi reads cost approximately 5 times more than PacBio CLR reads, and nearly 7 times more than Illumina reads [1,2,3]. The earlier TGS and NGS (Illumina) reads are cheap, however, and nearly every laboratory is equipped with these technologies. So, we are offering a solution “for everyone”: from the point of view of expenses, hybrid error correction (and also assembly) is a viable option for the majority of laboratories that have standard equipment at their disposal. As a

particular example, strain-aware analyses of pathogens become possible not only in an exclusive minority of sequencing labs, but in the vast majority of labs worldwide.

PacBio HiFi type and ONT Q20+ reads require to sequence a shorter template multiple times, and to generate a consensus from the multiple rounds of sequencing, as their core principle [1, 4]. This core principle, however, explains the inevitably increased costs, the shorter sequencing lengths, and, last but not least, also the significant computational investment for converting subreads into high quality consensus reads [1]. Also, although ONT Q20+ overall has very high accuracy on average, it suffers from large variability across individual reads (which may be generally unknown; and it is not obvious how to eliminate this variability by adjusting protocols). Hybrid correction is therefore still a helpful consideration when trying to improve the error content of the reads further [5]. As already briefly pointed out, the increasingly popular Ultra ONT platforms generate ultra long reads. These can span multiple repetitive regions to produce even longer consensus sequences. However, the error rate inherent to Ultra ONT reads is at >5%, where the spectrum of errors matches the one of other TGS reads (dominated by indel errors).

We have added additional comments/arguments with respect to these aspects to the manuscript.

References to this response:

[1] <https://www.nature.com/articles/s41576-020-0236-x>

[2] <https://www.nature.com/articles/s41576-022-00551-z#ref-CR140>

[3] <https://www.nature.com/articles/d41586-023-00512-4>

[4] <https://link.springer.com/article/10.1007/s40291-023-00669-8>

[5] <https://www.frontiersin.org/articles/10.3389/fmicb.2023.1043967/full>

Comment: Additionally, how does this compare to the approach of polishing the draft assemblies with NGS reads, i.e. running Racon directly on draft contigs to fix indel errors, which is also a common practice? Would it make sense to do a comparison here in this paper?

Response: Thanks for pointing out that a comparison with this practice is helpful guidance for practitioners. We answer this question together with the over correction you are mentioning further below.

As a protocol of the suggested practice, we first assemble raw reads using Canu, which is one of the most popular current long read assemblers, and then apply Racon to the resulting contigs. Results have been collected into Supplementary Table S5. Undeniably, Racon does correct many indel errors. However, if you look at mismatch error statistics, it becomes immediately evident that the amount of mismatch errors even increases, a clear indication of “overcorrection”. When applying HERO to Canu’s contigs instead, that is applying HERO for the purposes of polishing contigs as well (which is not the intended mode of application), the mismatch error content of the contigs assembled by Canu is further reduced. So despite not being intended, HERO works in the “polishing” scenario as well. Of course, when haplotypes are collapsed into consensus sequence prior to correcting errors (due to assembling raw reads), HERO can no longer help you either.

This last point becomes evident from comparing error content statistics with contigs resulting from the protocol that serves the purposes of HERO based assemblies, namely correcting errors first and only then assembling contigs from the corrected reads using

HiCanu (which is the better choice than Canu when corrected reads are available!). See Table 3, it becomes evident that this reflects the by far superior practice: the error rates drop substantially, Genome Fraction (which virtually measures haplotype awareness) increases and the misassembly rates drop, both significantly. See Table 3 and Supplementary Table S5 for those results.

Therefore, when striving for optimal results, we recommend to correct errors using HERO first, and assemble corrected reads only thereafter.

That does not mean of course that the (fairly common) practice you sketched, that is assembling raw reads and polishing thereafter, is justified for scenarios where one can agree on a handful of errors, because they do not disturb the intended downstream applications.

Comment: Large complex polyploid genomes may contain transposons which makes over-correction much more likely in the DBG step. Large metagenomics samples may be too difficult as shown in very high mis-assembled contigs, as measured in MC% in Table 3. The easier cases, on the other hand, may be straightforward whether you correct or not - so I really wonder if the method might just have a narrow window of applicability, or is it expected to work generally well outside the datasets tested in this work?

Response: One has to combine the assembly results displayed in Table 3 with the basic error correction statistics displayed in Table 2. It becomes immediately evident that HERO itself is not the issue: the reads that HERO delivers to the assembler are not affected by elevated MC rates [note that if MC rates (very slightly, if at all) increase after correction reflects evaluation artifacts, because HERO only corrects the errors, the reads themselves are neither touched in their basic sequence nor assembled into longer pieces, so there is no “assembly”, hence also no “misassembly”].

Only after application of the assembler to the reads corrected by HERO, the MC rates increase. This clarifies the situation: while HERO makes no mistakes, the assembler is obviously not able to pick up the haplotype/strain-aware corrected reads and assemble them into contigs correctly. If the assembler is not (fully) tuned towards processing strain-resolved reads, the assembler collapses reads from different strains into consensus contigs, which yields misassemblies (note that contigs aligning across two different strains during the evaluation are counted as misassemblies).

In summary, the issue is with Canu, the assembler that we had used, and not with HERO. Of note, it has already been demonstrated in earlier work that Canu does not necessarily output strain-resolved (or ploidy-resolved) assemblies (see e.g. results in [Luo et al., GB, 2021; Luo et al., GB, 2022]).

To provide further evidence of Canu not being tailored towards generating strain-resolved assemblies, we ran Hifiasm-Mate as an assembler alternative to Canu, see Table 4.

When running Hifiasm-Mate on HERO corrected reads, Genome Fraction significantly increases and MC rates drop significantly in comparison to Canu assemblies. Obviously, Hifiasm-Mate follows a protocol that can pick up the strain/haplotype-resolved input, and use it to the advantage of generating strain/haplotype-resolved assemblies.

In summary, HERO is evidently not the issue here.

Comment: To be honest, I was a bit concerned by the performance listed in Table 3. Specifically, the MC% (fraction of misassembled contigs) seem quite high for many of the HERO versions. Indeed, the HERO versions have significantly higher MC% compared to non-HERO versions in all cases presented. Given these misassembled contigs, I felt that the improvement on error rates and NGA50 isn't nearly as impressive as the authors claimed.

Response: As just pointed out, the increase in the MC rates when applying Canu to HERO corrected contigs is due to Canu not being able to work with the reads that have been error corrected in a strain-aware manner. So, Canu appears to be better provided with non haplotype-aware error corrected reads (which is a somewhat strange fix: curing a flaw with flawed input). Again, unlike Canu, Hifiasm-Mate is able to pick up the strain awareness that HERO delivers.

Beyond the Hifiasm-Mate runs, we provide additional evidence of assemblers to be the crucial factor in generating misassembly-free contigs. When having a look at results referring to the newly added datasets, *Arabidopsis thaliana* and *Arctia plantaginis*, which reflect trio (ancestor-offspring) data, it becomes evident that the switch error rate drops upon having corrected reads with HERO. In an evaluation scenario that lacks a reference genome, the switch error rate (as provided by Merqury) indicates how often contigs mistakenly switch phase, which is an immediate consequence of misassembled contigs. The switch error rate drops both in the reads themselves, and in the resulting assemblies, upon having corrected errors in the reads using HERO.

Comment: The component that is the most novel and deserves the most in-depth analysis is the step where the non-specific alignments are recognized and removed. This step is expected to reduce "over-corrections", which the authors described well in the Background, which is indeed a pain point for many prior methods. The important question here - are overcorrections fully addressed by HERO? It is not clear from the paper, and it would be helpful to use a few real examples to build up such an intuition and provide an estimated proportions of the overcorrections reduced due to this step.

Response: We thank you for this comment. In fact, we can provide evidence of overcorrection for Racon, which is based on an alignment based (earlier: OG based) strategy. Although not reflecting the original intention, we re-examine the contigs that are polished using Racon for which statistics are displayed in Supplementary Table S5, and compare Racon's with HERO's results. Evidently, the mismatch error rate increases after application of Racon, while it drops when applying HERO.

Both Racon and HERO adopt the strategy of aligning reads to contigs and then correcting the contigs (Supplementary Figure 4). However, HERO filters out reads from other strains, which avoids the phenomenon of over correction.

The bottom line is that Racon follows a protocol that permits to introduce new errors, or, in other words, to overcorrect sequences. HERO does never do this, insofar as the error rates *do not increase in any scenario* upon having HERO applied to correct the sequences.

To illustrate the phenomena, we use IGV and take Bmock12 PacBio data as an example to briefly describe over correction (Supplementary Figure 4).

Last but not least, we also corrected reads of four real datasets separately with Racon and HERO. We found that when the original mismatch error rate is not high, such as for Bmock12 ONT and Bmock12 PacBio, Racon increases the mismatch error rate in the corrected reads. HERO does not have this problem and significantly reduces both indel and mismatch errors.

Finally note that unlike for Racon, we are not able to provide evidence for DBG based hybrid approaches to overcorrect sequences by means of experiments. Nevertheless, we conjecture that this is one explanation for the error rates that are still considerably higher in comparison with the ones delivered by HERO. Note that the breaking down of reads into k-mers, which is at the core of DBG based methods, can disrupt the information of the co-occurrence of haplotype-specific variation (provably, as pointed out in plenty of earlier work!).

This means that variation can be confused across strains/haplotypes, which leads to long reads aligning with paths in the DBG that do not spell out haplotype/strain aware sequence. This has the potential for "overcorrection" as well.

We have amended all passages in the manuscript correspondingly.

Comment: Across the entire paper, it isn't clear which DBG implementation to use under which circumstances? The authors did not attempt to make recommendations among these choices. Are the users expected to try all three of FMLRC, LorDec, and Ratatosk? This is unclear. It might give the false impression that there is always some improvement in the HERO-version using any of the three methods. In real projects, the users unfortunately won't have a ground truth to compare to, so it often is quite difficult to know which DBG method to pick. As seen in Table 3, the winner can be any of F-, L-, or R-HERO and the authors should provide a rough guide.

Response: Thank you for this comment, we agree that recommendations like that would be very helpful for a practitioner. We recall that the primary purpose of this paper is to demonstrate that a combination of de Bruijn graph with alignment (earlier: OG) based procedures yields significant improvements in terms of correcting errors in TGS reads, with an additional view towards the quality of the downstream assemblies. Therefore, our response can hardly be conclusive in the frame of this manuscript. Your question appears to refer to a review style evaluation of protocols, which requires data sets that vary all in terms of ploidy (known or unknown) and also coverage statistics (including ratios TGS versus NGS coverage), and so on.

Note as well that an answer to your question does not depend so much on HERO, but rather on the choice of 'F', 'R' or 'L' itself, as the decisive criterion before integrating them into a HERO-style protocol. This becomes evident by studying results with a view on comparing 'F', 'R' and 'L' in their own right: in all cases, it is evident that they set the decisive mark, while HERO improves on the results they provide as a foundation. So, virtually, a study on providing recommendations on using F-, R- or L-HERO would turn into a study on providing recommendations on using FMLRC, Ratatosk or Lordec in the first place. We do not feel as if providing such advice makes a reasonable addition to this paper also for ethical reasons, because we are not the authors of these works.

Let us nevertheless provide the advice that we are able to provide. Based on an evaluation of our result sheets, we recommend to use F-HERO for metagenome data sets, and, more generally, for data sets in which the number and the relative coverages of the participating haplotypes are unknown. For genomes of known ploidy, where abundances of haplotypes are evenly distributed across haplotypes (on average along the genome), we recommend R-HERO.

We are confident to provide these recommendations also because of the general recommendations that guide Ratatosk and FMLRC in their original work: while Ratatosk evidently focuses on genomes of known ploidy, FMLRC puts decided emphasis on the treatment of metagenomes. According to our results, these qualities are preserved when applying the full R-HERO resp. F-HERO protocol.

We have put a brief comment on suggested application modes in the Discussion.

Comment: Another concern of mine is the time cost for HERO, which the authors acknowledged in Discussion - since there can be a few iterations and repetitions within each iteration- not to mention potentially run through all three DBG methods, F, L and R. To run the corrections for these many iterations/repetitions/methods may be prohibitive for larger metagenomics samples, or complex polyploid genomes. The authors did suggest that "TGS template based phasing procedure consumes the largest amount of time. A conceivable solution is to pre-phase the long TGS reads prior to aligning them with the NGS reads..." If

the authors already have an idea for performance improvement, why not just implement it to address this bottleneck?

Response: While we respect that your request would be favorable, we disagree on this point for a few reasons. Let us put it this way: “why not just implement it” is an innocent sounding statement. However, the corresponding claim is very demanding, and exceeds the limits of a revision. What we suggest as an outlook is not a matter of a quick fix of, say, one or two days. Rather, the pre-phasing of long reads refers to tedious analyses with respect to determining appropriate protocols and parameters, which to the best of our judgment would require at least a few weeks, if not a few months. So, extensive analyses are required to make it really run.

[From a larger perspective: isn't it rather uncommon to ask the authors to implement ideas for future work in a revised version of the project, in particular if that future work refers to details that do not have an influence on the core message of the paper (namely: combining DBG with alignment based procedures in hybrid assembly yields better results)? Also, as a thought experiment, what would happen if the implementation of the pre-phasing would trigger further future work...]

Because runtimes of R-, F- and L-HERO are not bad at all, we did not see an urgent need to implement related improvements immediately. The added alignment-based routine is fairly economic both in terms of runtime and peak memory.

Maybe there is also a misunderstanding: we do not recommend to run all of R-, L- and F-HERO. We only present all results to document that HERO leads to improvements on whatever choice of prior work you agree.

As for a recommendation, please see again above: we recommend to make use of F-HERO for datasets characterized by imbalanced coverage and unknown/uncertain ploidy (such as metagenomes) on the one hand, and R-HERO for datasets characterized by even coverage and known ploidy (such as polyploid plant genomes, for example). The recommendation for R-HERO is rooted in the superior results achieved in the category ‘Completeness’ in the corresponding tables.

Comment: "The POA can accumulate artifacts if not confined to small genomic regions." Please show evidence of this and how the segmentation bin size or boundary is determined.

Response: We use Racon's implementation for computing POA's, so follow Racon's reasoning. It is generally understood (basic sequence alignment knowledge) that the amount of artifacts in multiple alignments (and POA is a special type of multiple alignment) accumulates on increasing length of the sequences that make part of the multiple alignment. The evidence for this to happen should be documented in the literature in many places; Racon integrates such knowledge into its protocol, and overcomes the issue by segmenting the overall alignment region into smaller windows.

Comment: It may be my pet peeve, but the authors often use language in the paper that often sound more sophisticated than it really is. As an example, "As a technical detail, we computed POA's by making use of a single instruction / multiple data (SIMD) type of implementation ..." - it really is just invoking Racon - 'POA' and 'SIMD' is the implementation details of Racon - the authors should just spell out Racon in the methods.

Response: We agree (of course) that we call Racon (in combination with Minimap2), and we agree that the way we described it sounds as if we had come up with an implementation on our own, which is not correct. We apologize for putting down details in a misleading (and pretentious sounding) way. In the new version, we have amended that.

However, we indeed meant to highlight these technical details on purpose. The essence of our paper are methodical advances, new routines and protocols, and the idea of discussing technical details is to provide the interested reader with the methodical turns that appear to be crucial for our work. In other words, we did not select Racon by chance, but precisely for the reasons that we were trying to convey here: Racon enjoys an optimized and refined protocol for processing reads, which we explicitly appreciate.

Note that we have been concerned with related projects (haplotype-aware error correction and assembly) already for a while: see e.g. phasebook [Luo et al. Genome Biology, 2021], Strainline [Luo et al., Genome Biology, 2022], VeChat [Luo et al., Nat. Comms., 2022], StrainXpress [Kang et al., NAR, 2022]. We understand that the devil is in the technical details when trying to push approaches to higher limits. Therefore, we intended to communicate these technical, but (to the best of our understanding) crucial details.

Comment: Similarly, the use of overlap graph (OG) in HERO is quite trivial and a bit misleading since it doesn't traverse the graph structure as in a typical genome assembly algorithm. In the current context, it is just referring to sorting the overlaps suggested by MINIMAP2 into short read piles. There is no need for a graph data structure and nothing more than a sort or hash table with long read as the key.

Response: We apologize for not having communicated this clearly (our writing, quite apparently, was not illustrating the mechanics of our workflow sufficiently well). In any case, your description does not match what really happens in our workflow. In fact, there is an overlap graph, and it is even traversed to compute the error-corrected sequence of the long read that is to be corrected. Again, the devil is in the details hidden in the implementation of Racon. (another good reason to highlight its elements as guidance how real progress in correcting errors in long reads can be achieved)

Before providing an explanation, let us say that also in agreement with Reviewer 2 (see below), we return to referring to “overlap graph based approaches” as “alignment based approaches”, in agreement with prior work that followed this “nomenclature”.

So let us explain why our approach is overlap graph based indeed and, in fact we even traverse the overlap graph, to spell out the sequence that the traversal results in. To see this, one needs to look into the details of Racon. Racon relies on having reads aligned against a template (“backbone”) sequence. This agrees with what we do in Step 1 in Figure 1 in our work. Note that our Steps 1 and 2 do not agree with the procedures described in Racon, because Step 2, the phasing of the NGS reads, and the removal of unsuitable phases are based on our own ideas, so are novel (and decisive!). From Step 3, we make use of Racon. Let us have a look at what happens by citing the Racon publication:

“This alignment is needed only to split the reads into chunks that fall into particular nonoverlapping windows on the backbone sequence.”

This means that the alignments are not the basis for correction in Racon. The alignments only serve to provide coordinates along which one can split the NGS reads into chunks that are to be processed further by way of a multiple alignment.

Again, before Racon splits aligned reads into chunks, reads are filtered and artifacts are removed. Here, we make use of our own routines (Step 2 in Figure 1, further illustrated in Figure 2), which phase the reads. As a consequence, it enables us to successfully remove reads that stem from different phases (strains/haplotypes), which is crucial for strain/haplotype-aware error correction.

Then we start to make use of Racon. Citing its publication:

“Each window is then processed independently in a separate thread by constructing a POA graph using SIMD acceleration and calling the consensus of the window.”

This is a very brief description for a methodically involved routine (amply discussed in other places) In more detail, Racon (as do we, because we use it from here) constructs a POA graph from the NGS read segments, for each of the windows. Then, the consensus sequence is spelled out by the resulting POA graph.

The point now is that spelling out the consensus of a POA graph immediately translates into deriving an overlap graph from the POA graph and traversing it to compute an assembly. Importantly, deriving an overlap graph from a POA graph is an obvious procedure, by the definitions of POA and overlap graphs. See e.g. [Baajens et al., RECOMB 2020] for an implementation of this procedure that we provided by ourselves, and that we used earlier. The overlap graph that corresponds to the POA graph is then traversed to compute an assembly of the overlapping read chunks. This assembly is the consensus of the POA graph. The consensus of the POA graph then further reflects the sequence of the corrected long read in the particular window.

This explains why, although hidden in the details of Racon’s implementation, correcting reads virtually corresponds to constructing an overlap graph, from which the sequence of the corrected read is spelled out as the assembly that results from traversing the OG.

*Regardless of the details of the overlap graph that corresponds to the POA graph, the basis for correcting the TGS read are **not** the alignments of the NGS reads with the TGS read—the basis for the corrected sequence is an assembly of the NGS reads that does no longer depend on the alignments. And the basis for the assembly is, by no means, any *k*-mer based construction (such as a DBG). Also therefore it is correct to say that the corrected sequence results from an assembly that has an overlap graph as its basis.*

Note finally that we made use of related ideas also in earlier work of ours, beyond the one mentioned above (e.g. phasebook [Luo et al., GB, 2021]). This led us to categorizing HERO as “OG based”.

[On a side remark, note that we have been specializing in making use of overlap graphs for the purposes of enhanced phasing of reads already in plenty of earlier work, for example [Baaijens et al., Genome Research 2017] or [Gregor et al., Bioinformatics, 2016], which explains that we tend to identify overlap graphs wherever they arise, also if not explicitly pointed out.]

Last but not least, we do agree that our writing was not clear, because we did not explicitly highlight overlap graphs where they came up, even only in hidden form. In the revised version, we do this more explicitly. We also now refrain from calling tools “overlap graph based”, but stick to the old term “alignment based”. This of course is not wrong either, although it does not put emphasis on the competition in terms of paradigms between de Bruijn graphs on the one hand and overlap graphs on the other hand.

We have revised the manuscript wherever necessary to highlight these correspondences more clearly, wherever necessary, in particular in Results and in Methods.

Comment: Lastly, the code quality on GitHub is poorer than I'd expect and does not seem to fully match some of the methods described in the text. Please also consider including more examples and documentations.

Response: We have added more images and corresponding descriptions on GitHub. Due to GitHub's limits on upload data size, some examples have been directly uploaded to Code Ocean where readers can directly view or rerun them.
<https://codeocean.com/capsule/9666759/tree/v1>

Minor:

- Background appears too verbose and seems to contain substantial overlap between the initial section and Related Work section.

Response: It is our experience that the repetition of key statements helps to convey the take-home message to readers. This is certainly a matter of personal taste, so we hope for your understanding about our personal choices.

Nevertheless, we worked through the entire Intro and tried to wipe out any repetitions that create the impression of redundancy, much more than anything else.

- Explain R-Hero on its first occurrence (e.g. HERO corrected Ratatosk)
- Typo: "data structruess"

Response: Thank you for your feedback. We have added annotations and fixed the typos.

Reviewer #2:

Comment:

Overview: The authors consider the hybrid correction of long TGS reads using short and accurate short reads. This problem has previously been approached by e.g. aligning short reads to the long reads or by aligning long reads to the de Bruijn graph (DBG) of the short reads. Here the authors propose to combine these two approaches into a pipeline where first the DBG based approach is used and then the short read alignment based approach (the author use the term overlap graph (OG) approach) is used to further correct the reads. The authors implement this pipeline in a tool called HERO and show improvements over the previous methods when reads are corrected by HERO. The improvements are most pronounced in metagenomic settings where the different strains closely resemble each other.

Originally DBG based approaches replaced the alignment based approaches because they were superior when it came to space and time efficiency. I find it interesting that the authors show here that when leveraging the reads already corrected by DBG approaches and the highly efficient read mapper developed after the first correction attempts, the alignment based approaches become competitive also in terms of efficiency.

The paper is well written and easy to understand.

Response: We thank you for your positive evaluation.

Comment: Suggestions for revision:

1) The term alignment based approach has been used in previous literature to describe hybrid correction approaches relying on aligning short reads to the long reads. I do not see that it would be justified here to introduce a new term, OG based correction, for this approach.

Response: We thank you for pointing this out; in fact we had been overlooking the earlier "nomenclature". Let us nevertheless say that we are using overlap graphs, and say further that their use is crucial. The details are hidden in Racon's implementation, which we use for

processing the short reads in the windows that result from segmenting the alignment pile-up (see Step 4 in Figure 1 here).

Indeed, literally following the publication of Racon:

“This alignment is needed *only* to split the reads into chunks that fall into particular nonoverlapping windows on the backbone sequence.”

and further

“Each window is then processed independently in a separate thread by constructing a POA graph using SIMD acceleration and calling the consensus of the window.”

The latter description is very brief and spares the reader with some crucial technical details. The point is that constructing a POA graph and calling the consensus of the window based on the POA graph virtually corresponds to deriving an overlap graph from the POA graph, and traversing it to compute an assembly of the short read segments in that particular window.

In other words, this corresponds to a genuine overlap graph based procedure. Note that here we use Racon’s implementation to compute the consensus. However, we have made use of analogous procedures already earlier, using self-provided implementations of such practice (e.g. [Baaijens et al., RECOMB 2020; Luo et al., GB, 2021]), which may explain why the overlap graphs in these procedures become immediately obvious to us.

So, routines are both alignment and overlap graph based. As we do not intend to introduce confusion, we return to the precedents who agreed on categorizing such work as “alignment based”.

In the new version of the manuscript, we have applied the necessary changes and reformulations wherever applicable, in particular in Results and Methods.

Again: thanks for pointing this out.

Comment:

2) Page 7: "L-HERO ... its indel error rate amounted to only 37.9% of that of LoRDEC only:" I believe this should be 37.9% lower than LoRDEC, not 37.9% of the error rate of LoRDEC.

Response: Thank you for your thorough attention! We have applied the correction.

Comment:

3) Page 10: "HERO suppresses the indel and mismatch error rate by 60% and 20%, respectively, in comparison with the state of the art.": These results vary from one data set to another so this sentence should be reformulated to be more precise.

Response: We modified the sentence to: “HERO suppresses the indel and mismatch error rate by on average 65% (27~95%) and 20% (4~61%), respectively, in comparison with the state of the art.”

Comment:

4) Page 12: "we aligned all NGS reads on the one hand with all TGS reads on the other hand using Minimap2.": Unclear sentence, please reformulate.

Response: We modified the sentence to “We aligned each of the NGS reads on the one hand with each of the TGS reads on the other hand using Minimap2.”

Comment:

5) Some typos:

- Page 9: "...ONT data set, The performance"
- Page 10: "state-pf-the-art" (also just before state-of-the-art has been written without hyphens)
- Page 11: "data structurues"

Response: Thank you very much for your careful reading! We have corrected all of those typos.

Second round of review

Reviewer 1

I found lots of issues in the previous version - mostly regarding the clarity of the writing and explicit contributions around the technical improvements of HERO and RACON respectively, and challenged the authors to those claims. These concerns are largely addressed in the response and new revision, so it has become clearer to me what HERO's technical contribution is (and thanks to going through these with me).

One minor issue, I could not find reference for "Hifiasm-Mate", should it be "Hifiasm-Meta" instead?